# Reversible proliferative arrest induced by rapid depletion of RNase MRP

Yuan Liu[1,2,3,9], Shiyang He [1,2,3,9], Kawon Pyo [1,2,3], Reuben Franklin [1,2,3], Ibrahim B. Maaz[1,6], Chen Cai [1,7], Kriti Shah[1,8], Sihem Cheloufi [1,2,3] ✉, William F. Marzluff [4,5] ✉ & Jernej Murn [1,2,3] ✉

Cellular quiescence is a state of reversible proliferative arrest that plays essential roles in development, resistance to stress, aging, and longevity of organisms. Here we report that rapid depletion of RNase MRP, a deeply conserved RNA-based enzyme required for rRNA biosynthesis, induces a long-term yet reversible proliferative arrest in human cells. Severely compromised biogenesis of rRNAs along with acute transcriptional reprogramming precede a gradual decline of the critical cellular functions. Unexpectedly, many arresting cells show increased levels of histone mRNAs, which accumulate locally in the cytoplasm, and S-phase DNA amount. The ensuing proliferative arrest is entered from multiple stages of the cell cycle and can last for several weeks with uncompromised cell viability. Strikingly, restoring expression of RNase MRP leads to a complete reversal of the arrested state with resumed cell proliferation at the speed of control cells. We suggest that targeting rRNA biogenesis may provide a general strategy for rapid induction of a reversible proliferative arrest, with implications for understanding and manipulating cellular quiescence.

In a nutritive environment, proliferating cells invest most of their energy into growth and division[1]. However, under nutrient deprivation, single-celled organisms enter a state of quiescence, which substantially reduces their energy expenditure and secures their survival until nutrients again become available[2]. Cellular quiescence is also commonly used by multicelled organisms, not for survival of their individual cells, but for the benefit of the entire organism, including its development, long-term reproductive capacity, and tissue repair[3,4]. Despite the importance of quiescence in all domains of life, the mechanisms that mediate the transition between proliferation and quiescence remain incompletely understood.

Much of the interest in understanding the quiescent state is due to its essential role in maintaining adult stem cells and tissue functions over the lifespan of the organism[5]. Quiescent adult stem cells can reversibly exit the cell cycle to generate new differentiated cells, which are essential for tissue homeostasis and repair following injury. However, with aging or disease, the ability of adult stem cells to replenish missing or malfunctioning differentiated cells declines, contributing to the deterioration of tissue function[6–8]. This decline has spurred significant research into the molecular switches that regulate stem cell activation, with the aim of developing strategies to enhance tissue regeneration and combat aging[6,9]. Although progress has been made in

[1]Department of Biochemistry, University of California, Riverside, 3401 Watkins Drive, Boyce Hall, Riverside, CA, USA. [2]Center for RNA Biology and Medicine, 900 University Ave, Riverside, CA, USA. [3]Stem Cell Center, University of California, Riverside, 900 University Ave, Riverside, CA, USA. [4]Integrated Program for Biological and Genome Sciences, University of North Carolina, Chapel Hill, North Carolina, USA. [5]Department of Biochemistry and Biophysics, University of North Carolina, Chapel Hill, North Carolina, USA. [6]Present address: Department of Biochemistry, Stanford University School of Medicine, 279 Campus Drive, Beckman Center, Stanford, CA, USA. [7]Present address: Department of Human Genetics, University of Utah School of Medicine, 15 N 2030 E, Eccles Institute of Human Genetics, Salt Lake City, UT, USA. [8]Present address: Department of Microbiology & Immunology, School of Medicine, Stanford University, 299 Campus Drive, Sherman Fairchild Science Building, Stanford, CA, USA. [9]These authors contributed equally: Yuan Liu, Shiyang He. ✉e-mail: sihem.cheloufi@ucr.edu; william_marzluff@med.unc.edu; jernej.murn@ucr.edu

understanding how stem cells exit quiescence and re-enter the cell cycle, the mechanisms governing the transition from proliferation to quiescence remain less well understood—yet are equally crucial for both development and the long-term maintenance of tissue function[3,10,11].

A fundamental question is whether molecular mechanisms of cellular quiescence are conserved. The variety of quiescence-inducing signals (ranging from nutrient limitation in bacteria or yeast to altered availability of growth factors and cytokines in metazoans), entry points (largely from G1 but occasionally from other phases of the cell cycle), and durations (from a few days to several decades), suggests a diversity of regulatory schemes[2–4,12,13]. Yet, the fact that quiescence, as a cellular phenotype, is spread widely in phylogeny points to potential common underlying mechanisms that may have been evolutionarily adapted to different biological settings[3,14]. Defining such design principles will be key to understanding and manipulating cellular quiescence.

A common feature of quiescent cells is their remarkably low overall levels of ribosome biogenesis and protein biosynthesis, the latter being as low as 1% or less compared to proliferating cells[15–22]. This is notable because translation, including protein biosynthesis and ribosome biogenesis, is by far the largest consumer of energy during cell proliferation as well as a key driver of diverse cellular functions and growth[23–26]. A major reduction in the rate of translation can thus save a starving cell a substantial amount of energy to secure its survival. In fact, studies in bacteria and yeast demonstrate a universal engagement of active mechanisms to attenuate translation when nutrients become limiting[2,27–29]. It remains to be explored whether repression of energetically the most consuming processes, including ribosome biogenesis, itself serves as a strategy for induction of cellular quiescence.

Given their requirement for protein biosynthesis, tRNAs and rRNAs are fundamental determinants of a cell's energy economy and growth capacity[30–33]. Restricting the supply of rRNA or tRNAs slows cell growth, induces cell cycle arrest, and/or leads to cell death[34–38]. Conversely, increasing rRNA production is, remarkably, sufficient to accelerate proliferation of some already rapidly dividing immortalized cells[30,39–41]. Furthermore, it has been argued that a minimal level of rRNA is required for a quiescent cell to re-enter the cell cycle[42].

Biogenesis of nuclear-encoded tRNAs and rRNAs requires endonucleolytic cleavage by two related RNA-based enzymes, RNase P and RNase MRP[43]. RNase P removes the 5′ leader sequence from precursor tRNAs, and RNase MRP splits pre-rRNA into segments destined for incorporation into the small or large ribosomal subunits[43–45]. RNases P and MRP each comprise a unique catalytic RNA (ribozyme) and several protein subunits, most of which are common to both enzymes[46,47]. Since all the individual protein subunits are required for the enzymatic function in vivo[43,46,48–50], it follows that each of the common subunits are also required for the processing of both tRNAs and rRNAs. Little is known about the regulation of the levels or activity of RNases P and MRP or their subunits in cells.

Here, we investigate cellular and molecular consequences of rapidly depleting RNases P and MRP in immortalized human cells. Our results reveal a reversible, long-lasting proliferative cellular arrest, a phenotype triggered by rapid inhibition of rRNA biogenesis. We propose that inhibiting rRNA biogenesis may provide a general mechanism for the induction of a reversible proliferative arrest.

## Results

Previous attempts at inhibiting RNase P and/or MRP activities in mammalian cells have met with problems associated with inefficient depletion via RNA interference or low cell numbers after CRISPR/Cas9-mediated gene editing[51,52]. Given their essential cellular roles[43], we opted for a rapid depletion of human RNases P and MRP by directly targeting their protein subunits using a conditional degron tag approach (Fig. 1A and Supplementary Fig. 1A)[53–56]. Tagging just one

subunit is sufficient for enzymatic inhibition, since depletion of virtually any of the RNase P or MRP protein subunits reduces the levels of the associated catalytic RNA in cells[46,51,57].

### Inducible and rapid reduction of precursor tRNA and rRNA processing in human cells

Using homozygous knock-in of the FKBP12^F36V degron, we engineered HEK293T cells for targeted degradation of either the RNase P-specific protein subunit RPP21 (N21 cells) or a protein subunit that is shared between RNases P and MRP, RPP40 (C40 cells; Fig. 1A and Supplementary Fig. 1A, B)[43,55,56]. Treatment of N21 or C40 cells with a combination of cell-permeable degraders, dTAG-13 and dTAG^V-1 (henceforth dTAG)[55,56], led to potent and sustainable degradation of the targeted subunits within 1 h (Fig. 1B, C and Supplementary Fig. 1C–E).

Notably, dTAG treatment of C40 cells caused a rapid decrease in the levels of the catalytic RNAs of both RNase P (*RPPH1*) and RNase MRP (*RMRP*) whereas only *RPPH1* was depleted in dTAG-treated N21 cells (Fig. 1D–F and Supplementary Fig. 1F–H). These results are consistent with the specificity of RPP21 to RNase P and with the essential role of protein subunits stabilizing the catalytic RNAs[46,47,51,57]. A further inspection of the enzyme components revealed that loss of RPP40 led to co-depletion of all subunits making up the wrist and palm modules of RNase P, except for POP5 (Fig. 1A, G). In contrast, degradation of RPP21 did not affect the levels of any other protein subunit (Supplementary Fig. 1I). Since RPP40 directly interacts with all subunits of the palm module as well as the RPP29 subunit of the wrist module[58], we envision RPP40 as an interaction hub whose depletion leads to disintegration of a large portion of the RNase P and MRP complexes with subsequent destabilization of several protein subunits. Given that POP5 interacts with the stable finger module, we speculate that these interactions provide unique protection against degradation (Fig. 1A, G). In contrast, RPP21 is exclusive to the wrist module and only interacts with two subunits, RPP29 and RPP38. Its peripheral positioning within RNase P and limited protein-protein contacts may explain why RPP21 depletion does not affect the stability of other RNase P protein subunits (Fig. 1A)[58].

We next asked how the abruptly reduced levels of both catalytic RNAs in dTAG-treated C40 cells affected tRNA and rRNA processing. Northern blot analyses revealed a time-dependent accumulation of pre-tRNAs consistent with compromised cleavage of their 5′ leader sequences by RNase P (Fig. 1H), although the overall levels of stable mature tRNA did not change over 24 h. Similarly, there was rapid accumulation of 47S/45S rRNA and other RNA precursors consistent with a major defect in cleavage of the ITS1 segment by RNase MRP, blocking downstream rRNA processing of 18S, 5.8S, and 28S rRNA (Fig. 1I and Supplementary Fig. 1J). We observed no changes in the relative levels of 28S and 18S rRNA, in line with an equally important role of RNase MRP in the maturation of 28S and 18S rRNAs (Supplementary Fig. 1K).

To evaluate potential changes in total rRNA levels, we quantified 28S rRNA levels in C40 cells over time using fluorescence in situ hybridization-flow cytometry (FISH-Flow)[59]. We observed a 20–30% reduction in 28S rRNA levels at 2 days and a 50–60% reduction at 7 days of dTAG treatment (Supplementary Fig. S1L–N). We also observed a block in processing of tRNA but not rRNA precursors in dTAG-treated N21 cells, in agreement with depletion of RNase P but not MRP in these cells (Supplementary Fig. 1F, G, O, P). Together, these results establish a system for rapid, inducible ablation of tRNA and rRNA processing in human cells.

### Depletion of RNase MRP induces a long-term reversible proliferative arrest in human cells

Tight control of rRNA and tRNA biogenesis is critical for maintenance of cellular homeostasis and survival[30–32,34,35,60,61]. We considered how these parameters may be affected by an abrupt depletion of RNases P

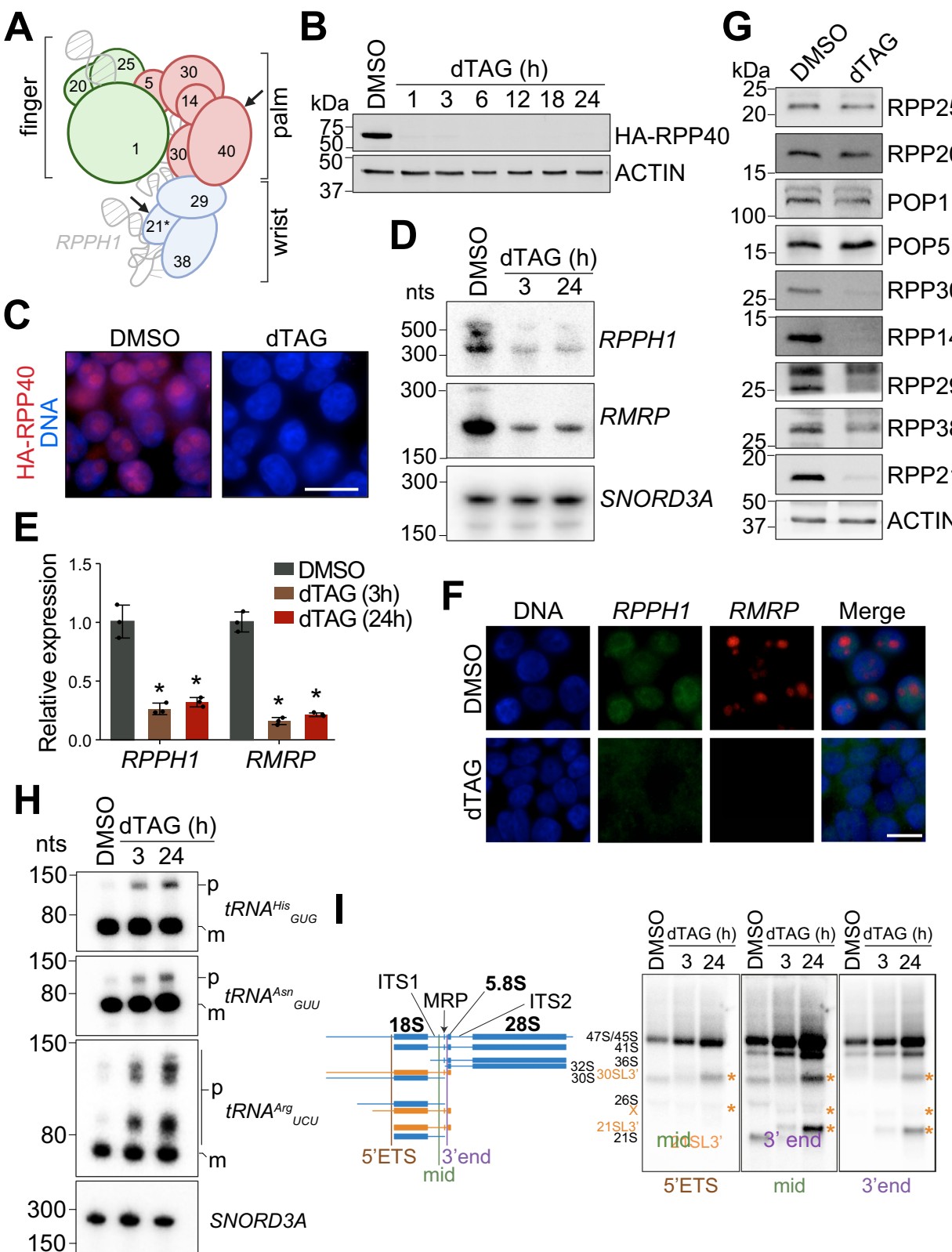

and MRP in human cells. We found that depletion of RNase P alone resulted in a reduced rate of cell proliferation starting at 5 days of dTAG treatment with little impact on cell viability (Supplementary Fig. 2A–C). In contrast, joint targeting of RNases P and MRP, while maintaining cell survival, caused a complete proliferative arrest between day 2 and 3 (Fig. 2A). The arrested C40 cells showed rounded morphology and weaker surface attachment compared to their vehicle-treated controls, but remained viable for several weeks (Fig. 2A, B). Strikingly, removal of dTAG after different times of treatment allowed the arrested cells to resume proliferation, with the delay before the proliferative restart correlating with the time of dTAG removal (Fig. 2A, C). The observed proliferative arrest and its reversibility are consistent with the defining properties of cellular quiescence as it occurs in primary cells in nature[4,62–66]. Furthermore, the relatively

**Fig. 1 | Inducible depletion of RNase P and RNase MRP in human cells.**
**A** Schematic of the human RNase P ribonucleoprotein based on the cryo-EM structure from Wu et al.[58]. Protein subunits RPP25, RPP20, POP1, POP5, RPP30, RPP14, RPP40, RPP29, RPP21, RPP38 are colored according to their position in the finger, palm, or wrist modules. The catalytic RNA *RPPH1* is in gray. Asterisk indicates the only known protein subunit that is not shared with RNase MRP, RPP21. Arrows indicate subunits targeted in this study. Created in BioRender. Murn, J. (2025) https://BioRender.com/qi2wxxp. **B** Immunoblot analysis of endogenous HA-tagged RPP40 in C40 cells treated with DMSO or dTAG for the indicated times. Actin serves as a loading control (*n* = 5). **C** Immunofluorescence of HA-tagged RPP40 in C40 cells treated with DMSO or dTAG for 3 h. Nuclei are stained with DAPI. Scale bar, 20 µm. **D** Northern analysis of *RPPH1* and *RMRP* from C40 cells treated with DMSO or dTAG for 3 h or 24 h (*n* = 3). *SNORD3A* serves as a loading control. A representative blot is shown. **E** Quantification by qPCR of RNA samples as in (**D**) (*n* = 3 biological replicates). Data represent mean ± SD. *, *p* < 0.01 (two-tailed Student's *t* test). Exact *p*-values are listed in the Source Data file. **F** RNA FISH of *RPPH1* and *RMRP* in C40 cells treated with DMSO or dTAG for 3 h. Scale bar, 20 µm. **G** Immunoblot analysis of RNase P and MRP protein subunits in lysates of C40 cells treated with DMSO or dTAG for 24 h (*n* = 3). **H** Northern blot of tRNA processing from samples in panel (**D**) using probes for the indicated tRNA genes. Primary $tRNA^{Arg}_{UCU}$ contains an intron removed separately from 5' and 3' sequences. m, mature tRNA; p, precursor tRNA. **I** Northern blot of rRNA precursors from samples in panel (**D**). Left, schematic showing positions of probes (5' ETS, mid, 3' end) and canonical (blue) and non-canonical (orange) pre-rRNA species[51,52,115]. RNase MRP cleavage site in ITS1 is indicated. Source data are provided as a Source Data file.

mild effect of inhibiting RNase P alone compared to the depletion of both enzymes suggests that the proliferative arrest is primarily induced by depletion of RNase MRP, although we cannot rule out a contributing effect of depleted RNase P.

We next asked how levels of the multiple RNase P and MRP subunits might correlate with the timing of the proliferative arrest and its reversal. Notably, unlike the rapid, dTAG-induced depletion of the enzymatic components (Fig. 1B–G), restoration of their levels upon dTAG removal required a considerably longer time (Fig. 2D, E). Interestingly, whereas cell division did not cease until several days of depletion of RNases P and MRP, resumed proliferation was observed roughly coincident with the re-appearance of the bulk of each enzyme (Fig. 2A, D, E).

To validate the idea that the restored levels of enzymatic components and resumed cellular proliferation indeed relied on expression of the genetically targeted RPP40, we derived a C40 cell line with doxycycline (Dox)-inducible expression of ectopic, Flag-tagged RPP40 (Fig. 2F). Despite a somewhat weaker depletion of RNases P and MRP in these cells, presumably due to the slight leakiness of the Dox-inducible cassette, dTAG treatment efficiently arrested their proliferation (Fig. 2F, G and Supplementary Fig. 2D). Importantly, addition of Dox during continued presence of dTAG promptly restored the levels of the enzymatic components as well as cellular proliferation (Fig. 2F, G and Supplementary Fig. 2D). We conclude that depletion of RNases P and MRP can cause cells to enter a reversible proliferative arrest.

We also determined whether other cultured cells could be induced into proliferative arrest in a similar manner. We engineered the colorectal carcinoma cells HCT116 for dTAG-dependent degradation of RPP40, as above (Supplementary Figs. 1A and 2E). Analogously to C40 cells, the modified HCT116 cells underwent a complete proliferative arrest upon dTAG treatment, remained viable, and resumed proliferation, though at a later time after washout compared to C40 cells (Supplementary Fig. 2F). Together, these results suggest that depletion of RNases P and MRP can induce a reversible proliferative arrest in human cells of different lineages.

Cells of various organisms predominantly enter quiescence from G1 phase of the cell cycle, although in certain settings quiescence can also be initiated from other cell cycle phases[63,67–69]. Our EdU incorporation assay in dTAG-treated C40 cells revealed a synchronous and time-dependent reduction in the rate of DNA synthesis, starting at 2 days of treatment; DNA synthesis rate was reduced to <10% at 4 days and completely ceased by 7 days of treatment (Fig. 2A, H, I). Strikingly, even at 7 days, a large fraction of cells, about 35%, contained a DNA amount typical of S-phase cells with a broad distribution between 2 C and 4 C DNA content (Fig. 2I, J). In contrast, the rate of DNA synthesis declined only moderately in dTAG-treated N21 cells, which did not arrest but continued proliferating with a slight increase in the proportion of cells in G1 phase over time (Supplementary Fig. 2G–I). Thus, driven by the depletion of RNase MRP, HEK293T cells enter a reversible proliferative arrest from multiple points in the cell cycle, with a substantial fraction of the arrested cells having an S-phase DNA content.

Perturbation of ribosome biogenesis through modulation of rRNA production can impact cell-cycle progression by stabilizing p53, which mediates the impaired ribosome biogenesis checkpoint (IRBC)[33]. To investigate whether the arrested state induced by rapid inhibition of rRNA processing was due to p53-dependent IRBC, we stably silenced the expression of p53 in C40 cells (C40^shP53) and compared their proliferation to control C40 cells (C40^shCTRL) during treatment with DMSO or dTAG, as well as after washout (Supplementary Fig. 3A). Despite undetectable p53 protein levels, dTAG treatment resulted in efficient inducible depletion of RPP40 and caused a complete proliferative arrest of C40^shP53 cells by day 3. Upon removal of dTAG at day 5, proliferation resumed about four days later (Supplementary Fig. 3B). In contrast to control cells, C40^shP53 cells exhibited a slightly delayed arrest and a moderately higher proliferation rate upon cell-cycle reentry, while viability remained high (> 80%) in both cell populations at all time points (Supplementary Fig. 3B). These results suggest that the cessation of cell proliferation induced by rapid depletion of RNase MRP occurs largely independent of p53-mediated IRBC.

To further characterize the observed proliferative arrest, we used immunoblotting to survey a number of key factors associated with p53-dependent or independent response to impaired ribosome biogenesis (p53, p21, HDM2)[70] and/or regulation of cell cycle (cyclin A, phosphoRb, E2F, PCNA, CDC6)[11]. We found that p53, whose protein levels increased only about two-fold over 7 days of dTAG treatment, was largely responsible for the induction of its downstream effector p21 (CDKN1A), although increased p21 was not necessary for the proliferative arrest or its reversibility (Supplementary Fig. 3B, C). We observed little change in the levels of other examined factors, including those downregulated as cells enter quiescence in G1, such as (phospho)RB or E2F, in line with the entry into the arrested state from multiple stages of the cell cycle (Fig. 2 and Supplementary Fig. 3C). We speculate that the mechanism of the induced reversible arrest may not exclusively rely on specific regulatory molecules, but could also involve a broader suppression of general cellular functions, which we explore in the following section.

### Induced decline in critical cellular functions

We investigated changes in other critical cellular functions as the HEK293T cells entered the proliferative arrest. Quantification of global translation via labeling of nascent polypeptides revealed a sustained rate in the first 24 h of depletion of RNases P and MRP, after which translation declined precipitously to less than 5% of the initial rate by day 7 (Fig. 3A, B).

Proliferating cells generally exert a higher metabolic rate compared to their quiescent counterparts[71]. To assess how depletion of RNases P and MRP might affect cellular metabolic activity, we measured the rate of ATP production by dTAG-treated C40 cells. Compared to the decline in translation, we observed milder reduction in total ATP production, which in the arrested cells after 7 days of treatment was reduced to 12% of the initial rate (Fig. 3C). We also found that the rates of ATP production via oxidative phosphorylation and glycolysis were comparably

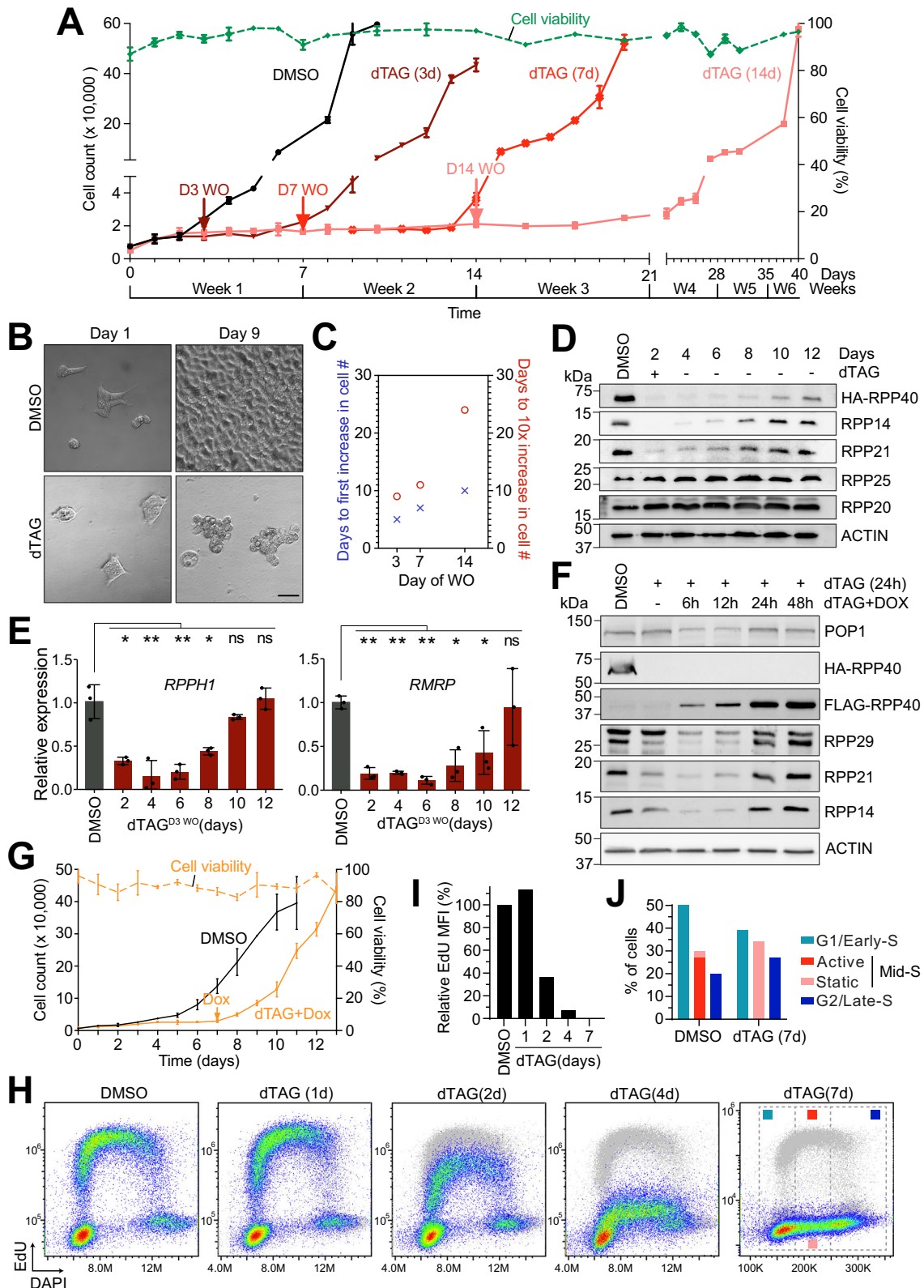

reduced, suggesting the cells maintained connectivity between the two main ATP-generating pathways (Fig. 3C). Furthermore, MitoTracker staining and measurement of mtDNA content showed evidence of reduced mitochondrial activity in the arresting cells, consistent with the reduced oxidative phosphorylation (Fig. 3D, E).

Quiescent cells must retain a basal transcriptional capacity to support important cellular functions necessary for survival[2,12,72–75]. We evaluated overall transcriptional activity of dTAG-treated C40 cells by labeling newly transcribed RNA with 5-ethynyluridine (EU) followed by click chemistry[76]. We found a surprisingly modest reduction in the rate of cellular RNA synthesis, which remained high at 65% at the time of a complete proliferative arrest 4 days after dTAG addition (Figs. 2A and 3F, G). Indeed, of the monitored cellular functions, the overall rate of transcription was the least affected, still at 17% after day 7 (Fig. 3G).

**Fig. 2 | Rapid depletion of RNase MRP induces a long-term reversible proliferative arrest. A** Growth of C40 cells treated continuously with DMSO (black) or dTAG for 3 (brown), 7 (red), or 14 days (pink), followed by dTAG washout (WO). Arrows mark the day of WO. The dashed green line shows viability during the 14-day dTAG treatment. Data are mean ± SD ($n = 3$ biological replicates). **B** Representative brightfield images of C40 cells on indicated days of DMSO or dTAG treatment. Scale bar, 35 μm. **C** Correlation between the timing of dTAG WO and time to first significant increase in cell number (blue) or 10-fold increase (red). Mean values from $n = 3$ experiments are shown. **D** Time-course immunoblot analysis of RNase P and MRP protein subunits in C40 cells treated with dTAG for 3 days before dTAG washout ($n = 3$). **E** qPCR of *RPPH1* and *RMRP* from RNA isolated as in (**D**) ($n = 3$ biological replicates). Mean ± SD. *$p < 0.05$; **$p < 0.01$ (two-tailed Student's *t* test); exact *p*-values in Source Data file. **F** Immunoblot analysis of RNase P and MRP

protein subunits, including the endogenously tagged RPP40 (HA-RPP40) and ectopic RPP40 (Flag-RPP40), in lysates of Dox-inducible C40 cells treated with DMSO or dTAG with or without Dox, as indicated ($n = 3$). **G** Growth of Dox-inducible C40 cells treated with DMSO (black) or dTAG, with Dox added from day 7 onward (orange). Arrow marks Dox addition. Dashed line shows cell viability. Mean ± SD ($n = 3$ biological replicates). **H** Flow cytometry of dTAG- or DMSO-treated cells after 1 h EdU pulse. EdU detected with Alexa Fluor 594; DNA stained with FxCycle Violet. In each dot plot, matching DMSO-treated control cells are shown in gray. **I** Relative mean fluorescence intensity (MFI) in mid-S phase cells over time. **J** Cell cycle phase distribution based on gating in (**H**) (dashed gray lines in day 7 plot). "Active" indicates cells synthesizing DNA, and "static" cells not synthesizing DNA. Source data are provided as a Source Data file.

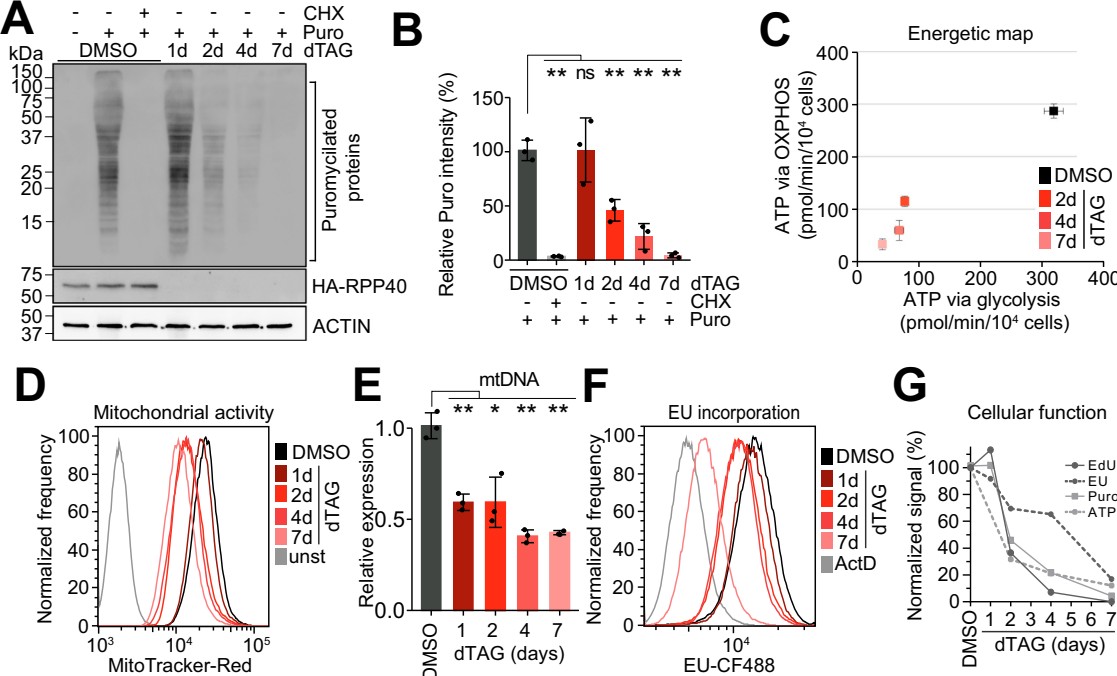

**Fig. 3 | Decrease of critical cellular functions. A, B** Decline in global translation. **A** Puromycin incorporation assay. C40 cells treated with dTAG, as indicated, were incubated with puromycin for 30 min. Cycloheximide (CHX) treatment was performed 10 min prior to puromycin and served as a negative control. Puromycilated proteins in cell lysates were detected by immunoblotting using an anti-puromycin antibody ($n = 3$). **B** Puromycin-incorporated protein levels in (*A*) were quantified and normalized to ACTIN levels. Data are shown as mean ± SD. *, $p < 0.01$; **, $p < 0.0005$ (two-tailed Student's t test); ns, not significant ($n = 3$ biological replicates). The exact *p*-values are listed in the Source Data file. **C–E** Reduced metabolic activity. **C** Reduced ATP production rate by dTAG-treated C40 cells. Control or dTAG-treated cells, as indicated, were analyzed by the ATP rate assay using a Seahorse analyzer (Agilent). ATP production rates through glycolysis or oxidative phosphorylation (OXPHOS) were determined as detailed in Methods. Data are

shown as mean ± SD ($n = 6$ biological replicates). **D** Decreased mitochondrial activity of dTAG-treated C40 cells, assessed by MitoTracker staining ($n = 2$). unst, unstained. **E** Decreased mitochondrial DNA (mtDNA) content of dTAG-treated C40 cells relative to controls. Measured by qPCR and normalized to genomic DNA. Data are shown as mean ± SD. *, $p = 0.0178$; **, $p < 0.01$ (two-tailed Student's *t* test) ($n = 3$ biological replicates). The exact *p*-values are listed in the Source Data file. **F** Reduced overall transcriptional activity. The rate of RNA synthesis in dTAG-treated C40 cells was measured by pulse-labeling with 5-ethynyluridine (EU) followed by flow cytometry. Actinomycin D (ActD)-treated cells served as a transcriptionally repressed control ($n = 3$). **G** Comparison of changes in key cellular functions during the transition of C40 cells from proliferation to dormancy. Time courses of average signals normalized to 100% for DMSO-treated cells from assays in (**A**, **C**, **D**, and **F**) are shown. Source data are provided as a Source Data file.

Collectively, these results document a coherent decline in critical cellular functions that leads to a reversible proliferative arrest in cells with very low levels of RNases P and MRP. The arrested DNA synthesis, minimal protein synthesis, and relatively higher metabolic and transcriptional activities suggest active maintenance of the non-proliferative but viable state.

## Acute transcriptional response centered on impairment of translation

To understand the initial cellular response to depletion of RNases P and MRP, we examined the transcriptomes of C40 cells at 3 or 24 h of treatment with dTAG by RNA-seq (Supplementary Fig. 4A). Of the

several thousand differentially expressed genes compared to control cells at each time, we found as the most prominently down-regulated a large group of factors implicated in differentiation and development, as well as genes linked to cell signaling, consistent with suppression of differentiation programs and reduction of cellular activity in quiescence (Fig. 4A, Supplementary Fig. 4B–D and Supplementary Data 1)[4,73,77].

Interestingly, many of the upregulated genes during the first 24 h would not be expected to promote a cessation of proliferation. These include genes essential for protein biosynthesis, including ribosome biogenesis factors, translation initiation and elongation factors, and regulators of tRNA processing (Fig. 4A, Supplementary Fig. 4C, D and

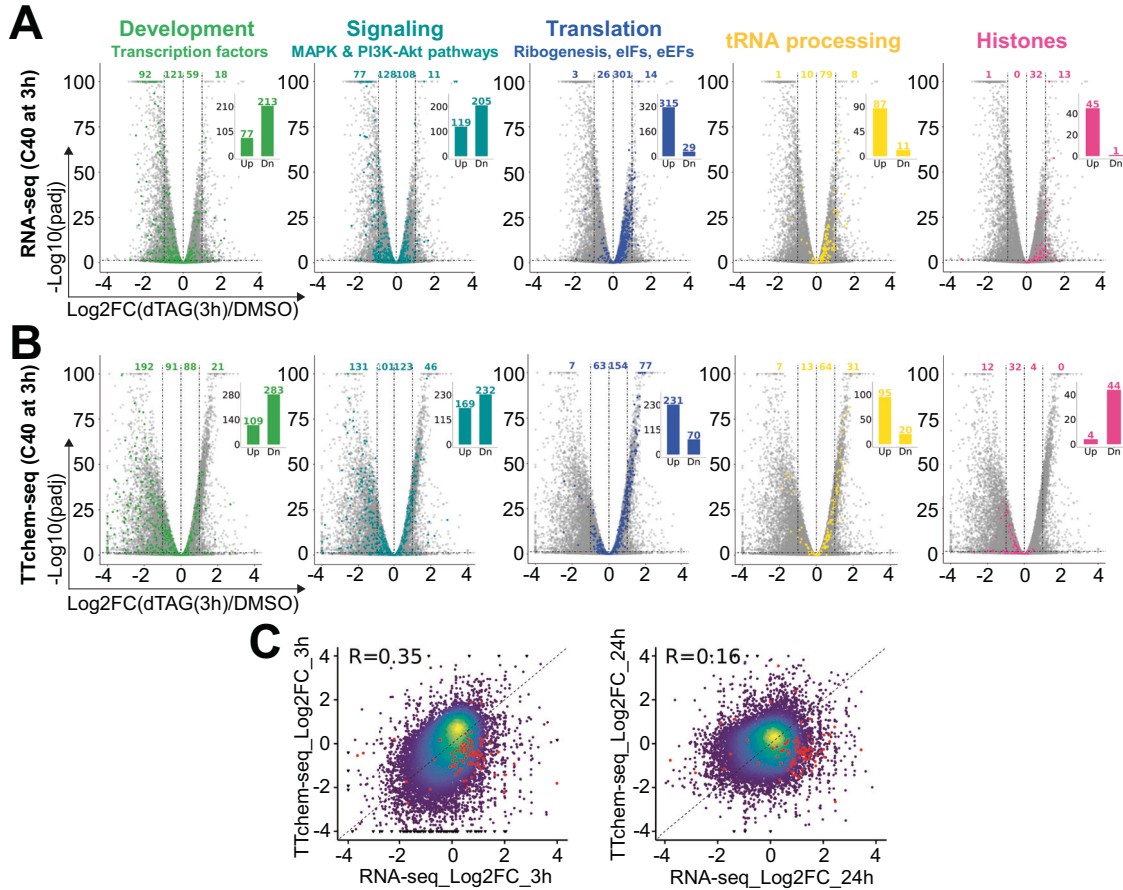

**Fig. 4 | Early transcriptional response to depletion of RNases P and MRP.**
**A** Impact of dTAG on steady-state mRNA levels. Volcano plots showing differential mRNA abundances between C40 cells treated with DMSO or dTAG for 3 h (RNA-seq data; $n = 3$). Gray dots indicate all evaluated genes, and colored dots genes belonging to the indicated functional categories (see also Supplementary Fig. 4B and Supplementary Data 1). Colored numbers above the volcano plots are counts of significantly regulated color-highlighted genes (adjusted $p$-value (padj) < 0.01) in different bins separated by dashed vertical lines according to the strength and sense of regulation. Insets summarize the total numbers of significantly up- or downregulated color-highlighted genes. FC, fold change. See also Supplementary Fig. 4 for changes at 24 h of depletion. **B** As in (**A**), showing the impact of dTAG treatment for 3 h on nascent transcripts ($TT_{chem}$-seq data; $n = 3$). **C** Correlation of RNA-seq and $TT_{chem}$-seq data at 3 h (left) and 24 h (right) of treatment. Red dots indicate significantly regulated histone genes (padj < 0.01). Differentially expressed genes in (**A**, **B**) were identified using the DESeq2 package. $P$-values were adjusted for multiple testing using the Benjamini-Hochberg method to control the false discovery rate.

Supplementary Data 1). The functional annotations of these genes suggest a direct cellular response to the acute perturbation of tRNA and rRNA processing, reminiscent of the 'countering' responses in microorganisms to quiescence-triggering lack of specific nutrients[78,79] or the response to impaired rRNA processing in zebrafish[80]. Curiously, among the most highly upregulated transcripts were histone mRNAs, which were not expected to accumulate in arresting cells that are not synthesizing DNA (Fig. 4A, Supplementary Fig. 4C–E and Supplementary Data 1).

To assess whether the observed changes in steady-state mRNA levels were driven by altered gene transcription or mRNA stability, we performed a global analysis of nascent transcripts using the $TT_{chem}$-seq protocol (Supplementary Fig. 4F)[81,82]. Despite the relatively poor overall correlation between RNA-seq and $TT_{chem}$-seq data, changes in transcription largely aligned with changes in steady-state mRNA levels for the most significantly enriched gene categories, particularly at 3 h of dTAG treatment (Fig. 4, Supplementary Fig. 4G and Supplementary Data 1). Strikingly, however, replication-dependent (RD) histone mRNAs showed an opposing pattern of regulation: their nascent levels decreased, indicating a decrease in transcription rate, but their steady-state levels increased, at 3 h and even more so at 24 h (Fig. 4 and Supplementary Fig. 4D, G), likely due to an increase in mRNA stability. Together, these results highlight transcription as a driver of

the most significant gene expression changes, with the notable exception of histone mRNAs, where posttranscriptional effects predominate.

We observed that the coherent induction of ribosome biogenesis factors, regulators of tRNA processing, translation initiation and elongation factors, and histones occurred upon combined depletion of RNases P and MRP, but not RNase P alone (Fig. 4 and Supplementary Fig. 4D–M), suggesting this was a major effect of RNase MRP. To directly gauge the contribution of RNase MRP, we examined the results of an earlier study by Goldfarb and Cech that sequenced total RNA of HeLa cells expressing Cas9 and CRISPR guides targeting *RMRP*[52]. Markedly, despite the composite cell population and relatively few reads mapping to mRNAs, the *RMRP*-targeted HeLa cells exhibited an overwhelming bias towards induction of the same highly regulated groups of genes as dTAG-treated C40 cells (Supplementary Fig. 4N, O). We conclude that much of the observed perturbation of the transcriptome, and the induction of the reversible proliferative arrest itself (Fig. 2 and Supplementary Fig. 2), is a result of the depletion of RNase MRP but not RNase P. However, given that depletion of RNase P alone moderately reduces the rate of EdU incorporation (Supplementary Fig. 2G, H), we cannot exclude the possibility that the co-depletion of RNase P, in combination with RNase MRP, contributes to the observed phenotypes in C40 cells.

### Histone mRNAs accumulate in proliferatively arresting cells

Surprisingly, the RD histone mRNAs were the most significant category of mRNAs whose levels increased by more than 2-fold in cells lacking RNases P and MRP (Fig. 4A, C, Supplementary Fig. 4D, E and Supplementary Data 1), despite the fact that the cells were not replicating DNA. They were also a prominent class of mRNAs whose steady-state levels increased after dTAG treatment despite decreased transcription. The RD histone mRNAs are normally expressed only in S phase, and their levels are coordinated with the rate of DNA replication. They also are the only cellular mRNAs in mammalian cells that are not polyadenylated, ending instead in a conserved stemloop that is a critical cis-element for their coordinate cell-cycle regulation[83,84]. Using qPCR, we confirmed the increased levels of histone mRNAs at 24 h and revealed their continued accumulation at longer times of dTAG treatment when cells essentially stopped proliferating or replicating DNA (Fig. 5A). We also found that restoration of RNases P and MRP following their depletion in the Dox-inducible C40 cells led to a decrease of histone mRNAs within 24 h back to their basal levels, demonstrating reversibility of the effect (Figs. 2F, G, 5B and Supplementary Fig. 2D). Fluorescence in situ hybridization showed that unlike normally cycling cells, in which histone mRNAs distribute throughout the cytoplasm (Supplementary Fig. 5A), in the arrested cells histone mRNAs concentrate in cytoplasmic puncta that become more numerous and distinct at longer times of dTAG treatment (Fig. 5C). We found that these puncta are different from stress granules (identified using antibodies against G3BP1, EIF4A, EIF4G, and PABPC)[85], which we did not detect in dTAG-treated C40 cells. These results suggest that when cells in S phase stop proliferating due to inhibited rRNA processing, their histone mRNAs may be stabilized by incorporation into storage granules and no longer translated.

The 3′ end of RD histone mRNAs is formed by endonucleolytic cleavage downstream of a conserved 3′ stemloop[83]. This cleavage requires SLBP, a protein that is also required for transport and translation of histone mRNAs and is present only in S phase[86,87]. In contrast, non-dividing, terminally differentiated cells lack SLBP but express a subset of the RD histone genes as polyadenylated mRNAs with extended 3′ UTRs[88]. Our RNA-seq and qPCR results show that the same histone mRNAs are expressed in the arresting cells as in control cells, and we found no evidence for expression of polyadenylated histone mRNAs in our RNA-seq data (Figs. 4A and 5A and Supplementary Fig. 4D). To determine whether depletion of RNases P and MRP might affect histone mRNA processing, we examined histone mRNAs in dTAG-treated C40 cells by northern blotting. This analysis confirmed the accumulation of histone mRNAs but showed no differences in their lengths, suggesting these mRNAs were processed normally (Fig. 5D).

Following transcription, histone mRNAs are cleaved 5 nts after the stemloop[83]. The mRNAs are then trimmed by a 3′ to 5′ exonuclease, 3′ hExo (ERI1), to leave a tail of 3 nts. The length of the tail is maintained at 3 nts by a terminal uridylyl transferase, TUT7, that adds uridines to the tail if it is shortened to < 3 nts (Supplementary Fig. 5B)[89]. In S phase, 30–50% of mature histone mRNAs contain nontemplated nts at their 3′ ends, which maintain the length of the tail and have no apparent effect on the stability or function of histone messages[90]. Typically, histone mRNAs are trimmed to give a tail of ACC, which is converted to ACU or AUU by cycles of 3′ hExo and TUT7 that shorten and then restore the length of the tail (Fig. 5E). As histone mRNAs get older, it is likely that the number of molecules that contain nontemplated termini will increase. We used EnD-seq to determine the 3′ terminal sequences of histone transcripts before and after RNase P and MRP depletion[90,91]. Analysis of H2A and H3 histone mRNAs confirmed that their 3′ ends were normally processed, but their nontemplated terminal uridine content increased two-fold after 24 h of RNase P and MRP depletion (Fig. 5F, G). This result is consistent with the histone mRNAs being older, which, in line with the transcriptome analyses (Fig. 4A, B and

Supplementary Fig. 4D, G), highlights their increased stability in cells lacking RNases P and MRP.

Upon completion or inhibition of DNA replication, polyribosome-associated histone mRNAs are rapidly degraded 3′ to 5′ in a process that requires SLBP and active translation[92,93]. SLBP protein is rapidly degraded upon completion of DNA replication, but not when DNA replication is inhibited in S phase[86,94]. Consistent with the finding that comparable proportions of normally cycling and reversibly arrested C40 cells contain a DNA amount typical of S-phase cells (about 30% and 35%, respectively; Fig. 2J), the dTAG treatment-induced cell cycle arrest did not alter the SLBP protein levels (Supplementary Fig. 5C). In contrast, repression of translation in the arrested cells would be expected to stabilize histone mRNAs, and low levels of their continued biosynthesis would result in their observed accumulation (Figs. 4 and 5A–D and Supplementary Fig. 4D, G). In line with this possibility, inhibition of DNA replication in C40 cells by a 30 min pulse with hydroxyurea (HU) caused only a small reduction in histone mRNA levels in translationally repressed, dTAG-treated cells compared to non-treated controls (Supplementary Fig. 5D). We suggest that compromised translation due to depletion of RNase MRP stabilizes histone mRNAs and leads to their cytoplasmic accumulation despite the reduced transcription and cessation of DNA replication.

## Discussion

Here we present evidence that acute depletion of a canonical component of RNases P and MRP causes loss of both ribozymes, disrupts biogenesis of tRNAs and rRNAs, and induces a long-term yet reversible proliferative arrest in mammalian cells. We propose a model where rapid inhibition of mature rRNA synthesis initiates a switch between proliferation and dormancy; when cells cannot sustain the minimum level of ribosome biogenesis that supports growth, a non-proliferative state is induced to preserve cell viability and function (Fig. 6). We speculate that this principle may apply both to single-celled organisms, in which quiescence is the default state under nutrient deprivation, and to cells of multicellular organisms where quiescence is induced by specialized, often cell type-specific triggers. Rapid depletion of RNase MRP induces a reversible proliferative arrest at multiple stages of the cell cycle, rather than allowing a cell to proceed through the cell cycle and arrest in G1, which often occurs when critical growth components are depleted slowly[2,63]. We suggest that inhibiting rRNA biogenesis with appropriate potency and speed may induce a reversible proliferative arrest in cells of different lineages. Prior observations supporting this possibility include inhibited proliferation but sustained viability of mammalian cells upon blockage of rRNA transcription[95] or processing[34,96,97], as well as induction of quiescence-like states in pluripotent cells by rapamycin (an mTOR inhibitor) or depleting Myc, which, among other effects, reduces the rate of rRNA biosynthesis[98–101].

The observed decline of major cellular processes, which includes a complete cessation of DNA replication and minimal protein synthesis, but a lesser reduction of ATP production and transcription, is reflective of the proliferative arrest accompanied by active maintenance of the dormant state. Notably, upon RPP40 depletion, the translation rate declines more rapidly than the availability of rRNA. For example, at 2 and 7 days of dTAG treatment, translation in C40 cells drops to below 50% and below 5% of the initial rate, respectively, outpacing the decline in 28S rRNA levels by approximately 2- and 10-fold (Fig. 3A and Supplementary Fig. 1N). This suggests that rRNA availability alone cannot fully explain the steep decline in translation. Although tRNA processing is also impaired, its contribution appears minimal, as depletion of RNase P alone does not significantly affect cell proliferation until day 6 of dTAG treatment (Fig. 1H and Supplementary Figs. 1O and 2A). We further consider it unlikely that the rapid translational decline results from the induced proliferative arrest. Instead, it is more plausible that the latter is a consequence of the

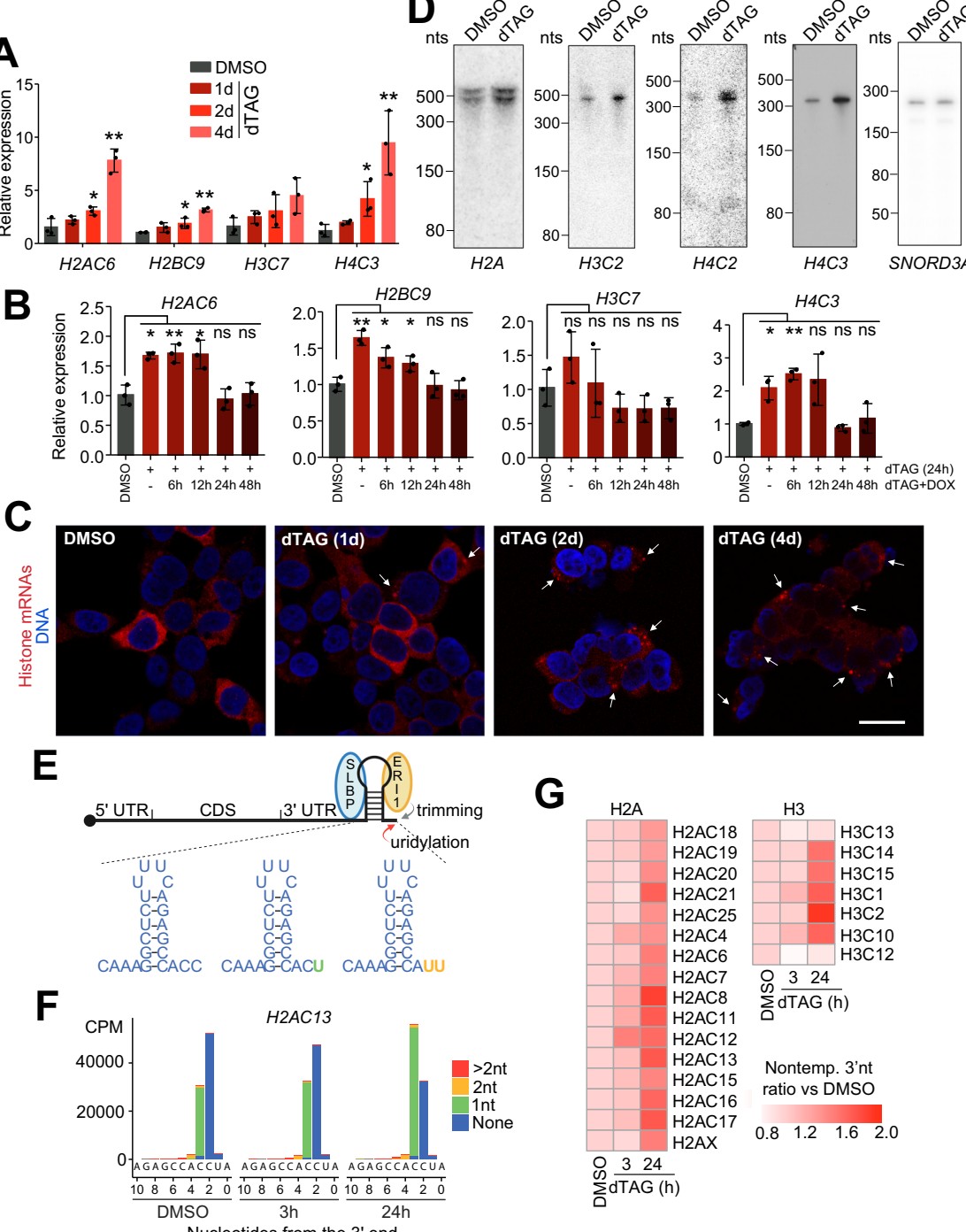

**Fig. 5 | Accumulation of histone mRNAs in cells lacking RNases P and MRP.**
**A** qPCR analysis of indicated histone mRNAs in control or dTAG-treated C40 cells
($n = 3$ biological replicates). Mean ± SD. *, $p < 0.05$; **, $p < 0.01$ (two-tailed Student's $t$
test). Exact $p$-values in the Source Data file. **B** qPCR of histone mRNAs in Dox-
inducible C40 cells treated with DMSO or dTAG, with or without Dox ($n = 3$ bio-
logical replicates). Mean ± SD. *, $p < 0.05$; **, $p < 0.01$ (two-tailed Student's $t$ test).
Exact $p$-values in the Source Data file. See also Fig. 2F, G and Supplementary Fig. 2D.
**C** Representative images of control and dTAG-treated C40 cells stained for DNA
(blue) and histone H3 and H4 mRNAs (red). Arrows point to cytoplasmic accu-
mulations of histone mRNAs. Scale bar, 20 μm. **D** Northern analysis of histone
mRNAs in C40 cells treated with DMSO or dTAG for 24 h. *SNORD3A* serves as a
loading control ($n = 3$). The *H2A* probes recognize multiple histone genes in the
H2A cluster. **E** Schematic of a histone mRNP with conserved 3′ stemloop bound by

SLBP and ERI1 (top). ERI1 trims 1–2 nucleotides, which are restored by uridylation
(green and orange U's), producing three distinct 3′ ends (bottom). Created in
BioRender. Murn, J. (2025) https://BioRender.com/1dxr6z2. **F** EnD-seq of *H2AC13*
mRNA from C40 cells after dTAG treatment. x-axis: last templated nucleotide; y-
axis: normalized read counts at each position (CPM, counts per million). Colors
indicate nontemplated tail lengths. The sequence below each plot indicates pro-
cessed histone *H2AC13* mRNA 3′ end formed in the nucleus; two nucleotides are
subsequently trimmed to yield cytoplasmic histone mRNA ($n = 3$). **G** Heatmap
summarizing EnD-seq data. Fold changes in nontemplated 3′ terminal nucleotide
content for dTAG-treated (3 h and 24 h) versus DMSO-treated control (0 h) con-
ditions are shown for each analyzed histone mRNA. Nontemplated nucleotides
were > 99% uridines. Source data are provided as a Source Data file.

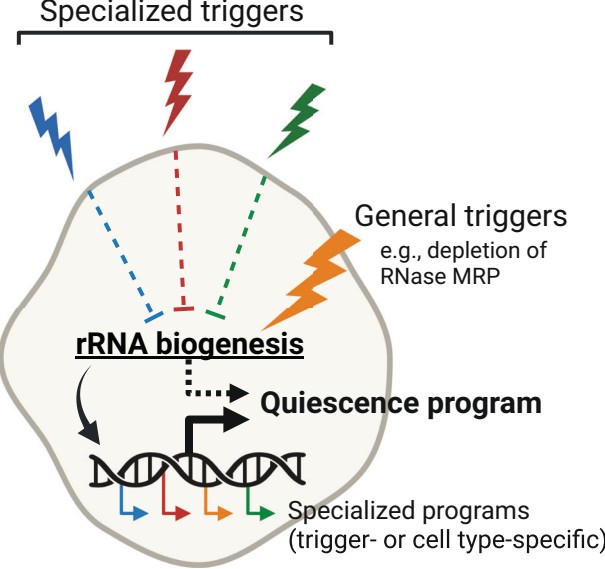

**Specialized triggers**

**General triggers**
e.g., depletion of
RNase MRP

**rRNA biogenesis**

**Quiescence program**

Specialized programs
(trigger- or cell type-specific)

**Proliferating cell**

**Fig. 6 | Model for induction of a reversible proliferative arrest via depletion of RNase MRP.** In primary cells, specialized triggers induce quiescence, in part, by attenuating rRNA (and thus ribosome) biogenesis indirectly, via signaling. In contrast, general triggers, such as rapid depletion of RNase MRP, induce a reversible proliferative arrest by directly blocking rRNA biogenesis, bypassing signaling. A 'quiescence program' is likely induced with involvement of changes in gene expression. Both specialized and general triggers of reversible non-proliferation also induce trigger-specific programs of gene expression that do not necessarily contribute to the induction of the arrested state. Created in BioRender. Murn, J. (2025) https://BioRender.com/nbroyt0.

reduced translation rate, as even a 50% reduction in translation (which we observe at day 2; Fig. 3A, B) is sufficient to induce cell cycle withdrawal (which occurs between days 2 and 3; Fig. 2A)[102]. Given that the rate of protein synthesis is limited by rRNA biogenesis[30], which is directly inhibited by depletion of RPP40, we propose that the primary driver of translational decline is disrupted rRNA biogenesis. We speculate that compromised rRNA biogenesis leads to translational decline through both reduced rRNA availability and signaling triggered by defects in rRNA processing. Further studies will be needed to dissect these mechanisms in greater detail.

Interestingly, the initiating translational block elicits a coherent induction of hundreds of genes that are required for translation. This suggests a trigger-specific response that is likely peripheral to the induction of the dormant state and that resembles responses of fission yeast to amino acid or phosphate starvation, which induces genes involved in amino acid or phosphate metabolism, respectively[78,79].

Global analyses indicate that much of the changes in gene activity resulting from the loss of RPP40 is due to depletion of RNase MRP rather than of RNase P, the former inducing a reversible proliferative arrest and the latter only slowing down cell growth. Our reanalysis of data reported by Goldfarb and Cech shows that similar changes in gene expression were triggered by rapid disruption of *RMRP* using CRISPR[52]. These changes included a coherent upregulation of genes required for translation and tRNA processing as well as histone mRNAs (Supplementary Fig. 4N, O)[52].

We also found that the striking accumulation of histone mRNAs in cytoplasmic granules of cells that no longer replicate DNA is a consequence of increased stability and a low level of continued expression of histone mRNAs, most likely restricted to cells that arrest in S phase. It is unclear whether the accumulated histone mRNAs facilitate reentry of the arrested cells into the cell cycle, but histone mRNAs return to

their normal levels rapidly upon dTAG removal, suggesting release from a translationally repressed state.

Although the rate of rRNA biosynthesis has long been considered a critical regulator of cell growth, the potency of its inhibition to induce cellular dormancy has not been appreciated. It is unlikely that rRNA biosynthesis is naturally shut off very rapidly in organisms, but it is conceivable that the speed and extent of its inhibition determine whether a cell continues to cycle or enters quiescence. This would automatically favor quiescence entry from the early G1 phase of the cell cycle, when the rRNA biosynthesis rate is low and only beginning to recover from a shutdown in mitosis[103].

## Methods
### Cell culture and treatment
HEK293T cells (ATCC, CRL-3216) and HCT116 cells (ATCC, CCL-247) were maintained in DMEM supplemented with 10% (v/v) FBS (Cytiva, SH30910.03) and 100 U/ml Penicillin-Streptomycin (Gibco, 15140122) at 37 °C and 5% (v/v) $CO_2$. To induce degron-mediated protein depletion, dTAG-13 (Tocris, 6605) and dTAG-V1 (Tocris, 6914) were dissolved in DMSO and used as an equimolar mixture at a final concentration of 500 nM (dTAG-13) and 1000 nM (dTAG-V1) for the indicated periods of time. To induce transgene expression, cells were treated with doxycycline (Dox; Millipore Sigma), dissolved in water, at 1 μg/ml. To inhibit DNA synthesis, cells were treated with hydroxyurea, dissolved in water, at a final concentration of 5 mM for 30 min.

To monitor the rate of cell growth, cells were seeded at 5000 cells per well in a 48-well plate. At each indicated time point, cells were harvested by digestion with 0.25% (w/v) trypsin-EDTA (Gibco, 2661762) from individual wells, and then counted using a hemocytometer (Fisher Scientific, 0267151B). Cell viability was determined via staining with Trypan Blue (Sigma, T8154-100ML). To wash out dTAG from cultures, cell culture supernatant was removed, attached cells were harvested by digestion with 0.25% (w/v) trypsin-EDTA, combined with the supernatant, and washed twice with complete media. Cells were then seeded into a new 48-well plate and counted at each indicated time point.

### Derivation of C40, N21, and HCT116 cell lines for dTAG-inducible protein degradation
Degron tag (FKBP^F36V)-containing knock-in cassettes were introduced homozygously into the last coding exon of the *RPP40* gene in HEK293T or HCT116 cells, yielding the C40 cell line or the RPP40-degradable HCT116 cell line, respectively, or into the first coding exon of the *RPP21* gene in HEK293T cells, yielding the N21 cell line, as previously described[56]. Modified plasmids pCRIS-PITChv2-BSD-dTAG (Addgene, 91792), pCRIS-PITChv2-dTAG-BSD (Addgene, 91795), and pX333 (for simultaneous delivery of the PITCh and gene-targeting gRNAs; Addgene, 64073) were introduced into the cells via transfection. See also Supplementary Fig. 1A. The Dox-inducible C40 cell line was generated via lentiviral infection of C40 cells, using the pLIX-403 plasmid (Addgene, 41395) to drive Dox-inducible expression of Flag-RPP40, as described previously[104]. The C40^shP53 and C40^shCTRL cell lines were derived by infection with lentivirus prepared from shp53 pLKO.1 puro (Addgene, 19119) and pLKO.1 Puro shRNA Scramble (Addgene, 162011), respectively, followed by selection with puromycin.

### SDS-PAGE and western blot analysis
Western blots were performed as described previously[104]. Briefly, whole cell lysates were run on 15% (w/v) SDS-polyacrylamide gels and transferred to supported nitrocellulose membrane (Bio-Rad) at 145 V for 1.5 h. Membranes were then blocked for 1 h in 5% (w/v) non-fat dry milk in TBST (137 mM NaCl, 20 mM Tris, pH 7.6, 0.1% (w/v) Tween-20), rinsed three times for 5 min with TBST, and incubated with primary antibody (1:1000 in 3% (w/v) BSA in TBST) overnight at 4 °C. Blots were washed in TBST 5 min for 3 times, incubated with HRP-conjugated

secondary antibodies in 5% (w/v) milk in TBST for 1 h and then washed again before developing. Beta-Actin was detected by incubating blots with HRP-conjugated beta-actin antibody (1:15,000 in 5% (w/v) milk in TBST) for 15 min followed by washing and developing. HRP signal was detected by developing with SuperSignal™ West Pico PLUS Chemiluminescent Substrate (Thermo Scientific, 34578).

## Immunofluorescence
Immunofluorescence was performed as previously described[104]. Briefly, cells were fixed in 4% (w/v) formaldehyde in PBS for 10 min at room temperature, permeabilized in 0.5% (v/v) Triton X-100 in PBS, blocked in 5% (v/v) FBS in PBS, and probed with primary antibodies at 4 °C for overnight. After an overnight incubation, the cells were probed with fluorochrome-conjugated secondary antibodies for 1 h at room temperature and mounted using VECTASHIELD Antifade Mounting Medium with DAPI (Vector Laboratories, H-1200-10). Images were taken with the LEICA DMi8 and LSM 880 confocal microscope (Zeiss), using LAS X (Leica) and ZEN (Zeiss) software.

## Antibodies
POP5 (sc-23046), RPP38 (sc-398113), RPP30 (sc-81374), RPP20 (sc-244043), RPP29 (sc-23048), and E2F1 (sc-251) antibodies were from Santa Cruz. RPP40 (Proteintech, 11736-1-AP), POP1 (Abcam, ab254978), RPP21 (Proteintech, 16377-1-AP), RPP14 (Invitrogen, PA5-57567), RPP25 (Sigma, HPA046900-100UL), SLBP (Invitrogen, PA5-66410), Histone H3 (Abcam, ab1791), FLAG M2 (Sigma, F1804-200UG), Puromycin (Millipore, MABE343), Cyclin A2 (Cell Signaling Technology,4656), pRb S807/811 (Cell Signaling Technology, 9308), Rb (BD Biosciences, 554136), P21(Cell Signaling Technology, 2947), PCNA(Cell Signaling Technology, 13110), CDC6 (Abcam, ab188423), and beta Actin (Sigma, A3854) were also used in this study. Mouse anti-p53/pab122 and mouse anti-MDM2/2A10 antibodies were gifts from Yanping Zhang.

## qPCR analysis
qPCR was performed as previously described[104]. Briefly, total RNA was extracted from samples using TRIzol Reagent (ThemoFisher, 15596018) and Direct-zol RNA Miniprep (Zymo Research, R2050) according to the manufacturer's instructions. cDNA was prepared from equal amounts of RNA using PrimeScript RT Reagent Kit (Takara, RR037A) following the manufacturer's instructions. qPCR was performed using PowerUp SYBR Green Master Mix (ThermoFisher, A25742) and the oligonucleotide primers listed in Supplementary Data 2 to amplify the cDNA on the CFX Connect Real-Time PCR Detection System, operated by the CFX Maestro (v2.2) software, at the annealing temperature of 63 °C. Relative mRNA levels were normalized to relative expression levels of the *SNORD3A* gene that was used as an internal control.

## Northern analysis
Northern blots with polyacrylamide gels were performed as described previously[52]. All northern probes, listed in Supplementary Data 2, were DNA oligonucleotides (ThermoFisher) 5′ end-labeled with $^{32}$P gamma-ATP (PerkinElmer, NEG502A250UC) and T4 polynucleotide kinase (New England Biolabs, M0201L) according to the manufacturer's instructions. Briefly, for visualization of RNAs <1000 nt in length (*RMRP*, *RPPH1*, *SNORD3A*, tRNAs and histone mRNAs), 5–10 μg of total RNA was separated on 10% (w/v) polyacrylamide/7 M urea/1 × TBE gels. RNA was transferred to Hybond N+ membranes (Cytiva, RPN203B) in 1 × TBE at 100 mA for 1 h at a cold room 4 °C and then cross-linked twice at 1200 mJ/cm2 before prehybridization in 10 mL of hybridization buffer (Invitrogen, AM8670) for 60 min at 42 °C. Probes were hybridized with the membrane overnight at 42 °C. The next day, the membrane was washed sequentially with the wash buffer 1 (2 x SSC, 0.1% (w/v) SDS), wash buffer 2 (1 x SSC, 0.1% (w/v) SDS), and wash buffer 3 (0.1 x SSC), once each for 30 min. The membrane was then

developed using the storage phosphor screen (GE Healthcare, 0146-931) in an autoradiography cassette (Fisher Scientific, FBAC810) overnight at − 20 °C. The next day, the phosphor screen was imaged with the Typhoon imager (Cytiva).

Northern blots with agarose/formaldehyde gels were performed as described previously[52]. Briefly, for high-molecular-weight RNAs, 10–20 μg of total RNA was heated for 5 min at 70 °C in 0.4 M formaldehyde/50% (v/v) formamide/1× TT (30 mM tricine, 30 mM triethanolamine)/0.5 mM EDTA, cooled to room temperature in a thermocycler, and separated on 1% (w/v) agarose/0.4 M formaldehyde/1× TT gels (15 cm × 15 cm × ~ 0.5 cm) in 1 × TT at 130 V for 5 min followed by 100 V for 3.5 h. The gels were rinsed with deionized water, soaked while gently shaking for 15 min in 50 mM NaOH, rinsed again with deionized water, and while gently shaking in 6 × SSC for 15 min. RNA was capillary-transferred to Hybond N+ membranes (Cytiva, RPN203B) in 6 × SSC overnight at room temperature. After transfer, membranes were cross-linked twice at 1200 mJ/cm² before prehybridization in 10 mL of the hybridization buffer (Invitrogen, AM8670) for 60 min at 42 °C. Probes (ITS1_3′end, ITS1_mid) were then hybridized with the membrane overnight at 42 °C. The next day, the membrane was washed sequentially with the wash buffer 1 (2 x SSC, 0.1% (w/v) SDS), wash buffer 2 (1 x SSC, 0.1% (w/v) SDS), and wash buffer 3 (0.1 x SSC), once each for 30 min. The membrane was developed using the storage phosphor screen (GE Healthcare, 0146-931) in an autoradiography cassette (Fisher Scientific, FBAC810) overnight at − 20 °C. The next day, the phosphor screen was imaged with the Typhoon imager (Cytiva) operated by the Amersham TYPHOON Control Software.

## 28S/18S ratio measurement
10–20 μg of total RNA was heated for 5 min at 70 °C in 0.4 M formaldehyde/50% formamide/1× TT (30 mM tricine, 30 mM triethanolamine)/0.5 mM EDTA, cooled to room temperature in a thermocycler, and separated on 1% agarose/0.4 M formaldehyde/1× TT gels (15 cm × 15 cm × ~ 0.5 cm) in 1 × TT at 130 V for 5 min followed by 100 V for 3.5 h. The gels were rinsed with deionized water, soaked while gently shaking for 15 min in 50 mM NaOH, rinsed again with deionized water, and while gently shaking in 6 × SSC with SYBR Gold (Invitrogen, S11494) for 15 min. Image acquisition was performed with a ChemiDOC imaging system (Bio-Rad) using Image Lab Touch 3.0 Software. ImageJ software was used to quantify the intensity of signals.

## RNA fluorescence in situ hybridization
RMRP-cy5, RPPH1-cy3 and ITS1-cy3 FISH probes were ordered from Integrated DNA Technologies. The custom Stellaris™ RNA FISH Probes were designed to target histone H3 and H4 clusters utilizing the Stellaris RNA FISH Probe Designer (LGC, Biosearch Technologies, Petaluma, CA) available online at www.biosearchtech.com/stellarisdesigner. All FISH probes are listed in Supplementary Data 2. The HEK293T C40 cells treated with DMSO or dTAG were hybridized with the Histone H3 and H4 Stellaris RNA FISH Probe set labeled with CAL Fluor Red 590 (Biosearch Technologies), following the manufacturer's instructions available online at www.biosearchtech.com/stellarisprotocols.

## Puromycin incorporation assay
To detect nascent polypeptides, C40 cells at 50% confluency were treated with 1 μg/mL puromycin for 30 min. As a negative control, CHX treatment was done at 10 μg/mL 10 min prior to adding puromycin. Cells were then harvested for SDS-PAGE and western blot analyses and blotted with the anti-puromycin antibody. Samples were tested in biological triplicates and image acquisition was performed with a ChemiDOC imaging system (Bio-Rad). ImageJ software was used to quantify the intensity of signals. Negative control lane's intensity was considered as background and was subtracted from the other lanes' intensity. The subtracted values were normalized to ACTIN and used for relative quantification.

## Flow cytometry

Samples were analyzed via flow cytometry by resuspending cells in FACS Buffer (5% (v/v) FBS, 1 mM EDTA, pH 8.0 in DPBS). Data was acquired with NovoExpress (v1.5) on NovoCyte 2100Y/Quanteon cytometers and analyzed with FlowJo (v9, v10). Cells were gated using forward and side scatter gating to exclude dead cells and debris, followed by height and area gating to exclude doublets. Cells were then analyzed as described below.

**Cell cycle analysis.** Cell cycle analysis was performed as described previously using the Click-iT Plus EdU kit (ThermoFisher, C10646)[105]. Briefly, at indicated times, dTAG-treated and control cells were pulse-labeled with 10 μM EdU for 1 h. One million cells were collected and fixed with 4% (w/v) formaldehyde and permeabilized with a saponin-based wash. EdU was labeled in a reaction cocktail with Alexa Fluor 594 or 647 picolyl azide for 30 min at room temperature. After labeling, total DNA content was stained with FxCycle Violet Stain (Invitrogen, F10347) in FACS Buffer for 30 min at room temperature and analyzed by flow cytometry. Cell cycle phases were determined based on DNA content and EdU signal. Relative mean fluorescence intensities (MFI) were calculated by subtracting the mean EdU intensity of unstained cells from the mean EdU intensity of stained cells in Mid-S phase at each time of dTAG treatment.

**Labeling of active mitochondria.** Mitochondria were labeled using the MitoTracker Red CMXRos (Invitrogen, M7512) following the manufacturer's recommendations. Briefly, after the indicated times, dTAG-treated and control cells were pulse-labeled with MitoTracker for 30 min. One million cells were collected and fixed with 4% (w/v) formaldehyde and permeabilized with a saponin-based wash. After permeabilization, DNA was stained with FxCycle Violet Stain (Invitrogen, F10347) in FACS Buffer for 30 min at room temperature before analysis by flow cytometry. Unstained cells were used as a negative control.

**Nascent RNA labeling.** dTAG- or DMSO-treated cells were pulse-labeled with 1 mM 5-ethynyluridine (5-EU) (Jena Bioscience, CLK-N002-10) for 30 min. One million cells were collected and fixed with 4% (w/v) formaldehyde and permeabilized with a saponin-based wash. Then, cells were incubated with a Click Cocktail (1 mM CuSO4, 43 mM Tris, 129 mM NaCl, 5.6 mM $C_6H_7NaO_6$, 50 μM CalFluor 488 Azide (Click Chemistry Tools, 1369-1)) for 30 min at room temperature in the dark. After incubation, DNA was stained with FxCycle Violet Stain (Invitrogen, F10347) in FACS Buffer for 30 min at room temperature before analysis by flow cytometry. Cells treated with Actinomycin D (2 μM) for 2 h were used as a negative control.

**Apoptosis labeling and detection.** Apoptosis was detected with Annexin V/SYTOX AADvanced kit (ThermoFisher, A35136). After cellular treatment, cells were trypsinized, collected, and washed with cold DPBS. Cells were resuspended at 1 million cells / mL in 1x annexin-binding buffer with 1:6 Annexin V and 4 μM SYTOX AADvanced dead cell stain. Cells were stained for 30 min at room temperature, then diluted 1:4 with 1x annexin-binding buffer and analyzed by flow cytometry. Cells with low Annexin V and SYTOX signal were considered healthy, cells with high Annexin V signal apoptotic, and cells that stained with SYTOX AADvanced dye were considered dead.

**Fluorescence in situ hybridization-flow cytometry (FISH-Flow).** FISH-Flow analysis of 28S rRNA was performed as described previously[59]. Briefly, dTAG- or DMSO-treated C40 cells were washed twice with PBS containing 2 mM EDTA. One million cells were fixed with 3.7 % (w/v) formaldehyde and incubated at room temperature for 10 min. Cells were then washed with PBS and resuspended in 600 μl of 70 % (v/v) ethanol. For staining, cells were transferred to a 96-well U-bottom plate and washed with 150 μl of FISH wash buffer (10 % (v/v)

formamide in 2xSSC buffer). Cells were then resuspended in 100 μl of hybridization buffer (10 % (w/v) Dextran sulfate, 10 % (v/v) formamide in 2X SSC) containing 1 μM FISH probes targeting human 28S rRNA (single-stranded DNA oligos 3' conjugated to Quasar 670; gift from Vikram R. Paralkar, University of Pennsylvania). The 96-well plate was sealed and incubated at 37 °C overnight. Cells were washed twice with 150 μl of FISH wash buffer and stained with FxCycle Violet Stain (Invitrogen, F10347) in FACS Buffer for 30 min at room temperature, then analyzed by flow cytometry. Relative mean fluorescence intensities (MFIs) of 28S rRNA were calculated by normalizing the geometric means of 28S rRNA-Quasar670 intensity to DMSO-treated cells.

A figure exemplifying the gating strategies is provided in the Supplementary Information (Supplementary Fig. 6).

## Mitochondrial DNA (mtDNA) quantification

Total DNA was isolated from dTAG- or DMSO-treated C40 cells using QIAamp DNA mini kit (Qiagen, 51304) according to the manufacturer's instructions. mtDNA was quantified by qPCR using primers amplifying the Cytochrome B (CYT-B) region on mtDNA relative to the beta-globin region on genomic DNA.

## Analysis of ATP production using Seahorse technology

Extracellular acidification rate (ECAR) and Oxygen consumption rate (OCR) were measured on a Seahorse XF Pro analyzer (Agilent) following the manufacturer's instructions. In brief, DMSO- or dTAG-treated C40 cells were seeded at $2.5 \times 10^4$ cells per well in Seahorse XFe96/XF Pro PDL plates. The next day, cells were equilibrated for 1 h in XF assay medium supplemented with 1 mM pyruvate, 2 mM glutamine, and 5 mM glucose in a non-CO2 incubator. OCR and ECAR were monitored at baseline and after the injections of Seahorse ATP Real-Time Rate Assay Kit reagents: Oligomycin (final concentration 1.5 μM) and Rotenone/Antimycin (final concentration 0.5 μM). Data were analyzed with Seahorse Wave Pro (10.1.0.1) software.

## RNA-seq analysis

Total RNA from aliquots of C40 or N21 cells treated with dTAG or DMSO was extracted using the Direct-zol RNA miniprep kit (Zymo Research, R2050) according to the manufacturer's instructions. Total RNA was subjected to one round of enrichment for poly(A)-containing RNA using the NEBNext Poly(A) mRNA Magnetic Isolation Module (New England Biolabs, E7490S), and sequencing libraries were prepared using the NEBNext Ultra II Directional RNA Library Prep with Sample Purification Beads (NEB E7765S). RNA-seq libraries were prepared, sequenced, and analyzed in triplicate. The libraries were subject to sequencing on Novaseq 6000, and at least 30 million reads were obtained for each sample. STAR (v2.7.3a)[106] was used to map the obtained sequencing reads to the human genome (hg38). Read counts in each gene (gencode v35) were calculated with the featureCounts tools (v2.0.0)[107] from the Rsubread package[108]. Differential expressed genes (DEGs) were identified with DESeq2 (v1.38.2)[109]. DEGs had the adjusted $p$-value < 0.05. RPKM values of genes were calculated with Cufflinks (v2.2.1)[110]. GO enrichment was performed with g.Profiler[111] with the significance threshold set to p < 0.01.

## Histone EnD-seq library preparation and data analysis

A detailed procedure was previously described[91]. Briefly, total RNA from DMSO- or dTAG-treated cells was extracted using the Direct-zol RNA miniprep kit (Zymo Research, R2050) according to the manufacturer's instructions. The RNA was then ligated to a linker oligonucleotide and reverse-transcribed to cDNA, based on the known 3' linker sequence. Six rounds of PCR were used to amplify several histone H2A or H3 genes using common, H2A- or H3-specific primers, listed in Supplementary Data 2, and a linker-specific primer. A second round of PCR was then performed using primers extended by the sequences of Illumina adapters. Histone EnD-seq libraries for each

sample were prepared, sequenced, and analyzed in triplicate. About 20 million 100-nt paired-end reads were obtained for each sample. Sequences of histone cDNAs were extended in silico 1 kb downstream from the 3′ ends to create a reference database for mapping. Reads of the 100-b paired-end sequencing data were mapped to the reference database with Bowtie2[112]. Umi-tools[113] was used to remove PCR duplicates. The nontemplated tails were identified with AppEnD[90].

### TT<sub>chem</sub>-seq library preparation and data analysis

Libraries for TT$_{chem}$-seq were prepared following a published protocol[82]. Briefly, C40 cells were treated with dTAG or DMSO and labeled with 1 mM 4SU for 10 min. Total RNA was isolated, fragmented, and biotinylated. The biotinylated RNA was enriched using μMACS™ Streptavidin Kit (Miltenyi Biotec, 130-133-282). After clean-up with RNeasy MinElute Cleanup Kit (Qiagen, 74204), NEBNext Ultra II Directional RNA Library Prep Kit (NEB, E7760S) was used to prepare Illumina sequencing libraries. The libraries were subject for sequencing on the Novaseq 6000 operated by the Real-Time Analysis software. About 70 million 100-b paired-end reads were obtained for each sample. The sequencing reads were mapped with STAR (v2.7.3a), and read counts for genes were calculated with the featureCounts tool (v2.0.0). DEGs were identified with DESeq2 (v1.38.2). DEGs had the adjusted $p$-value < 0.05.

### Reporting summary

Further information on research design is available in the Nature Portfolio Reporting Summary linked to this article.

## Data availability

The high-throughput sequencing data generated in this study have been deposited in the Gene Expression Omnibus (GEO) database under accession number GSE253620. Source data are provided in this paper.

## Code availability

The original code used to analyze the data and generate figures is available at https://github.com/Shiyang-He/Cellular-quiescence-data-analysis/ or at Zenodo: https://doi.org/10.5281/zenodo.15354217[114].

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

## Acknowledgements

We thank Claire Armstrong, Fabio Cerignoli, Mary Hamer, David Carter, and Melanie Oakes for helpful discussions and assistance with experiments, and Vikram Paralkar for generously providing the 28S rRNA FISH probes used in this study. This work was supported in part by grants from the US National Science Foundation (2054195) to J.M. and S.C., and the National Institutes of Health (GM29832) to W.F.M., S.H., and R.F. were supported by a TRANSCEND fellowship from the California Institute of Regenerative Medicine (Award #EDUC4-12752). J.M. was supported by the U.S. National Institutes of Health grant R01 GM144693. S.C. was supported by the U.S. National Institutes of Health grant R35 GM151004.

## Author contributions

Y.L., S.H., W.F.M., and J.M. conceptualized the study and designed experiments and data analyses. Y.L., S.H., K.P., R.F., I.M., C.C., and K.S. performed all experiments. S.H. performed bioinformatic analyses. S.C. contributed key reagents and provided guidance. S.C., W.F.M., and J.M. acquired funding and supervised this work. J.M. wrote the manuscript with assistance from W.F.M. and input from all other authors.

## Competing interests

J.M., W.F.M., and S.C. are inventors on a patent application filed by The Regents of the University of California (U.S. Provisional Patent Application No. 63/581,832) describing the methods of inducing reversible proliferative arrest presented in the manuscript. J.M. and S.C. are the founders of StopTime Inc., a startup focused on identifying and developing methods and compounds for the induction of cell dormancy and hypometabolism. All authors declare no other competing interests.
