## [Transparent Peer Review file · Nature Communications]

Reversible proliferative arrest induced by rapid depletion of RNase MRP

Corresponding Author: Dr Jernej Murn

Editorial Note: Figure R1 in this Peer Review File has been amended to remove third-party material where no permission to publish could be obtained.

Version 0:

Reviewer comments:

Reviewer #1

(Remarks to the Author)

The authors use an array of cutting-edge approaches to demonstrate that targeted degradation of human RPP40, a protein subunit associated with RNase P and RNase MRP, triggers cellular quiescence. This central finding is of great interest to biology and medicine. Despite a large body of data and the work's significance, strengthening the study/manuscript will help bolster the mainstay. To lend cogency to the narrative, certain intriguing observations and seemingly inconsistent findings also warrant elaboration. During revisions, the authors should consider the following comments/questions.

1. The statement in the Introduction that "there is little known about the mechanisms that mediate the transition between proliferation and quiescence" should be expanded appropriately. This nexus has been of longstanding interest (albeit not with a focus on RNase MRP) given the multifaceted ramifications of quiescence. For example, Adam et al. (1987) *Exp. Cell Res.* 169: 345-356 showed that quiescent mammalian fibroblasts could be stimulated to enter the cell cycle provided that an rRNA threshold was fulfilled. Moreover, there has been considerable effort by many investigators to understand the molecular switches essential for regulating wide-ranging processes during the activation of quiescent stem cells, and during developmental versus cellular quiescence. Thus, the Introduction would benefit from a brief consideration of the linkages between quiescence and stem cells/aging, and a more focused lens on specific knowledge gaps pertinent to this study.
2. The authors cite Goldfarb and Cech (2017) *Genes Dev.* 31: 59-71, who used the CRISPR/Cas system to generate RNase MRP-depleted strains (ref. 33). Goldfarb and Cech demonstrated that loss of the RNase MRP RNA results in proliferation arrest of HeLa cells. Since Goldfarb and Cech showed that heterogeneity in RNase MRP depletion was likely because some cells escaped either transfection or CRISPR cleavage, it is reasonable that authors of this study favored the inducible and reversible degron approach to deplete a protein subunit of RNase MRP. However, given that RPP40 is associated with RNase P and RNase MRP, the observed effects on quiescence cannot be assigned solely to depletion of RNase MRP, notwithstanding the comparison between depletion of RPP21 (N21) versus RPP40 (C40). The text should indicate clearly that a combined effect of decreased RNase MRP and RNase P function cannot be ruled out (see #8 and #24 below). The pre-tRNA processing defect is more evident in C40 than N21 cells; the EdU assay to assess cellular proliferation shows that there is an effect even in N21 cells.
3. Given the above caveat and the availability of many *Saccharomyces cerevisiae* RNase MRP mutants, including mutations in proteins that are associated with only RNase MRP and not RNase P, it seems that the quiescence trigger could have been unambiguously established in yeast. Was there a compelling reason to not leverage these mutant alleles?
4. Although RNase MRP RNA and protein levels are almost completely depleted in C40 cells, there remains some "non-canonical" processing of 47S/45S precursor rRNA. Is there another enzyme that is compensating for RNase MRP loss, albeit inefficiently/inaccurately? In addition, the pattern of non-canonical rRNA maturation products observed here is different from that observed when the MRP RNA subunit was depleted (ref. 33). While it is useful to show these rRNA intermediates, what are the levels of 18S and 28S rRNA? Are mature rRNA levels altered at all? Please include these data.
5. Fig. 1D and Fig. 1E: The northern blot shows that RNase MRP RNA is nearly absent, yet the quantitation shows that both the 3-h and 24-h time-point have one-fourth or one-third (of the control) remaining. Are the relative expression levels also normalized based on the loading control?

6. The ectopic inducible expression of RPP40 to rescue the degron-mediated disruption of native RPP40 is an appealing approach. However, it is not clear why the kinetics of rescue are so different between the dTAG washout and the Dox-inducible system. Does the degron system display a delayed homeostatic reset (post-washout)? If this is the case, please cite appropriate literature.

7. The northern blot data for the dTAG+Dox condition must be included (i.e., primary data for Fig. S2C) because it is intriguing that RNase P and RNase MRP RNAs are recovering even before expression of FLAG-RPP40 (6-h timepoint). Or, does the variable limit of detection of these different measurements render a one-to-one correlation hard to establish? Please explain.

8. RNase P has been depleted using the degron system to dampen the levels of either RPP40 (C40) or RPP21 (N21), two different protein subunits of human RNase P. There are some inexplicable findings that merit mention and perhaps even some speculation as to the underlying bases. First, it is surprising that the levels of select mature tRNAs either stay the same or increase slightly (but not decrease!) in C40 and N21 cells (Fig. 1H and Fig. S1J); the finding is particularly notable given the pronounced down-regulation of the RNase P RNA as well as several protein subunits in the C40 cells (compare Figs. 1D and S1F). It is possible that pre-tRNAs are subject to a higher turnover rate than their mature counterparts and the northern blot underestimates the 5' biogenesis defect. Worth including this perspective? Second, the precursor-tRNA (pre-tRNA) accumulation appears to be identity dependent. For instance, tRNA^{Arg}(UCU) does not seem to be affected at all in C40 cells in stark contrast to N21 cells. Why? Third, in the case of N21, only RPP21 is depleted. The other RNase P proteins (RPPs) seem largely unaffected. In C40 cells, however, the depletion of RPP40 results in loss of RPP21, RPP29, RPP38, RPP14, and RPP30. If the loss of H1 RNA is due to exposure of a naked RNA (uncoated with cognate protein subunits) to nucleases, one would have predicted a more severe reduction in C40 cells but this is not the case. So, what accounts for similar changes in H1 RNA levels in N21 and C40 cells? Are the relative levels of RNase P RNA versus RNase MRP RNA transcripts known? Does RNase MRP RNA somehow compensate for the loss of RNase P RNA and stabilize the free protein subunits? Fourth, POP5 and RPP14 each form a heterodimer with RPP30. It is not easy to appreciate why depletion of RPP40 would affect RPP14 and RPP30 but not POP5. Please explain/speculate. Fifth, why is POP1 barely detected in C40 cells, even in DMSO controls, when it is readily detected at levels comparable to other protein subunits in N21 cells. Last, human RNase P RNA has multiple bands. Why? Fig. S1F also shows more than one species, but these appear to be smaller (unlike Fig. 1D).

9. Some of the RNA-seq data are puzzling. For instance, despite pre-tRNA build-up in both N21 and C40 cells, changes in the expression of tRNA processing-associated genes seem to be different (Fig. 4 and Fig. S3). What about tRNA transcription? Any data?

10. Histone mRNAs are among the most highly upregulated transcripts upon depletion of RPP40. What about the levels of histone proteins (or even PTMs)? How does the increase in histone mRNAs correlate to cell cycle arrest? What happens to the levels of histone proteins upon entry into or exit from quiescence? The link is not clear. Moreover, why do the histone mRNAs form cytoplasmic punctae (Fig. 5C)? Are other RNAs also present in these bodies and are these punctae reversible liquid condensates? Based on the explanation provided for the 3'-processing of histone mRNAs, will incubation of the cytoplasmic extracts of dTAG-treated cells result in non-templated nucleotide addition? Additional experimentation with histones is clearly necessary to build a complete picture. Thus, although the histone angle is interesting, the 3' processing aspect is a distraction from the RNase MRP mainstay. Clearly, direct versus indirect effects that ensue from downregulation of RNase MRP need to be parsed, but such an investigation is beyond the scope of this study. Best to focus on the theme that RNase MRP is one switch to toggle between quiescence and proliferation.

Other comments

11. Text and figure edit (global): 5' not 5'; 3' not 3'.

12. The Introduction ends rather abruptly. This section would benefit from a final short paragraph summarizing the goal(s) of the study, the key findings, and the main takeaways.

13. TTchem-seq should be defined and briefly described as readers are less likely to be familiar with this relatively recently developed method compared to traditional RNA-seq.

14. Fig. 1A: Protein subunits not drawn to scale (e.g., compare POP5 versus RPP14). Why?

15. Fig 1F: Quite useful. Why is an equivalent not provided in Fig. S1?

16. Fig. 1G (and elsewhere): Complete western blots (i.e., corresponding to the entire gel that was probed) should be included in the supplement. At least one representative blot for each protein tested. This comment is motivated by the fact that the western blot for RPP29 shows two bands and only one of these bands is affected post-dTAG treatment (Fig. 1G). In fact, the situation is even less clear in Fig. 2D where there is cyclical change in the intensity of one of these two bands. Likewise, Fig. 2D shows two bands for RPP20. Significance?

17. Fig. 1G, line 109: While an RNase P/MRP aficionado might understand this comment on finger-to-wrist assembly pathway, it would be helpful to include a better explanation based on the findings of Engelke, Krasilnikov, and co-workers.

18. Fig 1H and Fig S1J: The tRNAs should read GUG (not GTG), GUU (not GTT), and UCU (not TCT).

19. Fig. 1I: Labels do not appear to be consistent with the annotated bands. What does the orange-colored asterisk indicate? What is meant by "orange diagrams" in the legend? Binding sites of 'mid' and '3' end' probes are not clear from this figure. The solid line-format in Goldfarb and Cech (2017) may be easier for the reader.

20. Fig. 2A: Please change the color of the line indicating "cell viability" as it likely to be confused with some other similarly-colored lines in the same panel.

21. Fig. 2C: The y-axis labels are not easy to comprehend. Please revise. Also, it is not clear why cells that have entered quiescence would respond differently in terms of the day of washout?

22. Fig. 2D and Line 142: What is meant by "bulk of each enzyme"?

23. Fig. 2H: The range of the x-axis in the dTAG (14 day) panel is narrower than in all of the other panels. This trend is not replicated to the same extent in the N21 cells (Fig. S2F). Rationale?
24. Fig. 2I and Fig. S2G: What is the interpretation here? Do these data not support the contention that there is a combined effect of RNase P and RNase MRP while examining the EdU data?
25. Fig. 2J and line 298: How does fig 2J show that the distribution of C40 cells in S phase is either 30% normally cycling or 35% quiescent? Unclear.
26. Fig. S1J: Is the additional band for pre-tRNA^{Arg} because of an intron-containing intermediate?
27. Fig. 3: Many upregulated transcripts in C40 correspond to genes involved in tRNA processing and translation. How does the transcription rate of tRNAs change?
28. Fig. 4C: Color ramp key?
29. Fig. 5E: Lacks the final two nucleotides (UA) in Fig. 5F. Fix.
30. Fig. 5F: Explain the colors and bar graphs. Unclear what is the message with respect to uridylation.
31. Fig. 6: The arrow and right-hand part are moot. Delete. (The figure itself does not add value.)
32. Line 78: Please include a citation.
33. Line 174: Replace "downturn" with "inhibition" or another suitable word.
34. Line 246: What is meant by "highly induced"?
35. Line 252: Please include a citation.
36. Line 274: Are the figures cited here correct?
37. Line 276 (and elsewhere): Southern is a proper noun (thus, Southern blot), but northern is not.
38. Methods: (a) Please use SI units for time (h, min, s). (b) Specify % in either v/v or w/v.
39. References: Titles need to be uniformly formatted in sentence case (not title case).
40. Welting et al. (2006) RNA 12: 1373-1382 should be included in the list of citations, as this paper showed the association of RPP40 with RNase MRP.

Reviewer #2

(Remarks to the Author)

Reviewer #3

(Remarks to the Author)

Reviewer #4

(Remarks to the Author)

MRP impacts on rRNA biogenesis and induces a quiescence cell cycle state. This is accompanied by acute transcriptional reprogramming and a decline in critical cellular functions. Interestingly, the authors also observed a strong increase in the levels of histone mRNAs, which accumulate locally in the cytoplasm. Altogether, the findings are of interest, but overall remain at a rather descriptive level. In particular, it remains unclear whether and how the induction of histone mRNAs is connected to the quiescence phenotype. Further, it is long established that impaired ribosome biogenesis impacts on cell-cycle progression by activating a p53-dependent impaired ribosome biogenesis checkpoint (IRBC). The authors need to dissect to what extent the quiescence phenotype is connected to the canonical IRBC and why cells enter G0 phase. In this context they need to compare p53-deficient and proficient cells and additionally should explore the contribution of the pRB-axis. Furthermore, the authors need to validate whether their observation can be expanded to depletion of other regulators of rRNA biogenesis. Use of an RNA polymerase I inhibitor and depletion of a set of other rRNA processing factors would be needed for generalizing their observation. Without providing more mechanistic insight and generalizing their observation, the findings are of limited significance.

Reviewer #5

(Remarks to the Author)

In this manuscript, Liu and colleagues investigate the effects of depleting RNase MRP in human cells. MRP splits pre-rRNA into segments that will be incorporated into small or large ribosomal subunits. After this enzyme is depleted, there is a decrease in the synthesis of ribosomal RNAs. The cells stop cycling and accumulate histone messenger RNAs. The cells stop cycling at different stages of the cell cycle and can remain in this state for weeks and alive. Restoring expression of MRP results in resumed proliferation at the speed of control cells.

Comments

Overall

The rapid depletion of human RNases P and MRP with the conditional degron approach provides for a first understanding of the impact of removing these two proteins. The ability to wash out the degradation inducing dTAGs represents an advantage for these studies because it is possible to test the reversibility of these effects.

If the authors would like to argue that depleting RNase P and MRP induces cells into quiescence, it would be better to perform these studies in cells that enter a quiescent state in physiological conditions, for instance, primary cultures and stem cell models. The 293 cells used here are so far removed from being in a body and likely have so many cell cycle changes that they are not a good choice for studies of quiescence. HCT116 is also a poor choice because cancer cells do not become quiescent the way that primary human cells do.

A cell that is not proliferating isn't necessarily quiescent. If the authors argue that loss of these RNaseP/MRP is sufficient to make cells quiescent, the authors should characterize the state they have entered to see if signature hallmarks of quiescence are present. Johnson and Cook lay out characteristics of quiescent cells (Fac Rev 2023 12: 5). These are listed in Table 1 as reduced cyclin expression, increased expression of CDK inhibitors, low CDK activity, unlicensed DNA replication origins, low RB phosphorylation, and others. The authors should choose a model system in which the cells can become quiescent *in vivo* and test the impact of depleting RNase P and MRP in this system. They can also perform RNA seq on RNaseP and MRP depleting cells, proliferating cells, and cells induced into quiescence by different signals and determine whether the RNase P/MRP depletion results in gene expression changes that are similar to the relevant quiescence signals for the chosen cell type.

In their abstract, the authors argue that targeting rRNA biogenesis is a general mechanism for the induction of cellular quiescence. To make this claim, the authors should show that in at least their chosen quiescence model, and for this claim, multiple models would be needed, that physiologically relevant quiescence signals result in decreased rRNA biogenesis and changes similar to that observed in their MRP degradation experiments. Ideally, the authors would then show that if they apply a relevant quiescence signal, but prevent the cells from downregulating rRNA biogenesis, that the cells fail to enter a proper quiescent state.

Specific comments

1. The authors state that all quiescent cells have remarkably low protein biosynthesis rates but only cite a review in yeast. Such a statement would need more validation across many quiescent systems.
2. The authors say that it remains to be explored whether reducing rRNA biogenesis. There is a substantial literature on rapamycin which reduces rRNA biogenesis and has been studied extensively as a mechanism for inducing a cell cycle arrested state.
3. In Fig 1, the authors show an effect of the ablation on tRNA and rRNA. What happens to mRNAs?
4. Fig 2B. Those cells look sickly, possibly like they are apoptosing. A viability dye may not pick up early apoptosis. Quiescence is not usually associated with a visible rounding and detachment from the plate.
5. In Fig 2H and J, the authors show that the cells stop in every cell cycle phase upon depletion of RNaseP/MRP. It is true that some examples of cells entering quiescence in phases other than early G1 have been discovered. But it is still typical for cells to enter quiescence in early G1. The authors should perform these experiments in cells that enter quiescence in response to physiological signals, treat those cells with quiescence inducing signals or inactivate RNaseP/MRP and determine whether the cell cycle phase distribution produced is similar or different. If quiescence signals induce the cells to exit in G1 and RNaseP/MRP inactivation results in a range of cell cycle states, this would not support the authors' conclusion that rRNA biogenesis inactivation is the common trigger for quiescence.
6. The authors observe a large induction of transcripts for ribosome biogenesis, translation initiation and elongation, regulators of tRNA processing, and histone mRNAs when RNaseP/MRP are inactivated. If ribosomal biogenesis inactivation is a common mechanism of quiescence induction, are these same gene categories induced when cells enter quiescence?
7. What is the authors' model for why histone mRNAs accumulate with RNase P/MRP knockdown? Why are histone mRNAs more stable but other transcripts expressed in S phase are not?
8. In the last sentence, the authors speculate that in early G1, rRNA biosynthesis is naturally low, so that is why many cells exit the proliferative cell cycle in G1. Is there any evidence to support this argument? There are cells that are reported to enter quiescence from other cell cycle phases. Is there any evidence that they have low ribosomal biogenesis at an unusual time in the cell cycle that's driving this? Can the authors test this by increasing ribosomal biogenesis and determine whether cells that normally quiesce now do so at a different phase of the cell cycle?

Version 1:

Reviewer comments:

Reviewer #1

(Remarks to the Author)

The authors have strengthened the manuscript with various revisions. As stated in a review of the previous version, the authors' central finding that downregulation of RPP40, a protein subunit associated with RNase P and RNase MRP, triggers cellular quiescence is of great interest to biology and medicine. Thus, the work warrants publication. However, the authors are encouraged to consider the following comments/questions during the next (final?) round of revisions.

1. In response to a previous critique that it would be desirable to document the levels of 18S and 28S rRNAs post-depletion of RPP40, the authors now show these data in Supplementary Figure 1K. Moreover, they comment in the text that the levels of these rRNA do not change upon RPP40 depletion. Given these new data, it is intriguing how depletion of RNase MRP leads to quiescence. While it is possible that ribosomes fully assembled before RPP40 (and RNase MRP) depletion have long half-lives and are not turned over rapidly, the effects on proliferation from RPP40 down-regulation are rather immediate. The authors must describe these findings and provide a reasonable explanation, as this critical gap is related to the mainstay.

2. In many instances, it would be beneficial for the information provided in the rebuttal to be included in the revised manuscript (after addressing some additional perspectives listed below).

(a) It is evident that depletion of RPP21 and RPP40 leads to inactivation of RNase P in N21 and C40 cells, respectively. Moreover, there is a similar reduction in H1 RNA levels even though the protein subunits are affected rather differently. In the rebuttal and revised text, the authors mention that H1 RNA level would plummet if the holoenzyme were not fully assembled. This speculation is reasonable. However, the explanation provided for the lack of POP5 degradation seems inadequate. If contacts between POP5 and other RPPs in the finger module of RNase P account for protection of this protein but not RPP14, it is intriguing that the same argument does not hold in N21 cells. Overall, the interesting differences in the RPP levels in C40 versus N21 cells (Figure 1G versus Supplementary Figure 1I) would benefit from a stronger rationale.

(b) It is fine that the non-canonical pre-rRNA processing products accumulate when RNase MRP is depleted, but where do these products come from? Is it mis-cleavage by partially-assembled, weakly-active MRP? Dysregulation of specific endo/exonucleases? RNA self-cleavage?

(c) The slower moving bands in the H1 RNA northern blot are being compared to the scenario in the Akiyama et al. (2022) bioRxiv. However, this earlier paper only shows bands that migrate faster compared to the putative full-length H1 RNA species. Thus, Akiyama et al. (2022) bioRxiv does not seem an ideal reference. Perhaps, the slower RNA is a circularized version?

(d) New data are provided with respect to tRNA transcription in the rebuttal (Fig R7) but not in the revised manuscript. The number of tRNA transcripts impacted in N21 cells, however, is significantly lower than in C40 cells. Is there any reason why this is the case? Regardless, please include this figure in the revised supplement and include a comment in the text.

(e) "Our results demonstrate a coherent transcriptional induction of genes associated with translation and tRNA processing in C40 but not N21 cells..." – Figure S4 shows there is still significant upregulation of translation and tRNA processing associated genes in N21 cells, at least at the 3-h time point, but not to the same extent as in C40 cells. At 24 h, these genes are significantly downregulated. Is there an explanation for reversal of this trend?

3. To address a previous critique pertaining to the striking differences in accumulation of pre-tRNAs, the authors have repeated the northern blot for tRNAArg (new panel in Figure 1H, which is very different from the previous data). It is unclear if the loading control for this new blot is also being used as the reference for the older northern blots related to tRNAHis and tRNAAsn. If so, the top two panels and their corresponding loading control (previous version of the manuscript) should be presented together, while the new tRNAArg blot should have its own loading control. Since the loading control (SRNORD3A) data in the previous versus current submission are different, the revised Figure 1H appears to be a mélange. If so, please fix.

4. In their rebuttal, the authors state "The faint band that appears at a higher-than-predicted molecular weight (MW) in the RPP20 blot in Fig. 2D is likely non-specific, while both bands in the RPP29 blots in Figs. 1G and 2F are likely specific, as their intensities change coherently." The latter comment does not seem to be accurate. In Figure 2F, it is obvious that the intensity of the two main RPP29 bands pre- and post-Dox induction are reversed. Perhaps, they constitute isoforms that arose from splicing or post-translational differences. Worth pointing out.

Minor points

1. Consider revising the title to "Cellular quiescence is induced upon rapid depletion of RNase MRP".

2. In response to the comment that the RPPs should be drawn to scale in Figure 1A, the authors stated that they followed the representation PMID 30454648. In Figure 5A of PMID 30454648, however, POP5 and RPP14 are similar in size. Therefore, for the sake of accuracy, the schematic in Figure 1A should be revised.

3. Interesting that RNase MRP appears to localize to speckles in the N21 cells compared to the larger structures in C40 cells. Comment?

4. Could the authors speculate why S-phase arrest due to MRP depletion does not result in histone mRNA degradation as was observed when S-phase arrest is triggered by replication inhibition?

5. Pg 5, ln 134: The data are not consistent with "a substantial accumulation". Please revise text.

6. Pg 5, ln 169: Revise to read "To validate the idea that...."

7. Pg 5, ln 172: Revised to read "...due to the slight..."

8. References: As indicated before, titles need to be uniformly formatted in sentence and not title case (including but not limited to references 2, 4, 18, 33, 58, 61).

Reviewer #2

(Remarks to the Author)

Reviewer #3

(Remarks to the Author)

Reviewer #4

(Remarks to the Author)

In their revised version the authors have strengthened the manuscript by addressing the p53-dependency of the quiescence phenotype and by adding data using the RNA polymerase I inhibitor. Expanding the link between quiescence and rRNA biogenesis by depleting additional biogenesis factors would still have been important.

To my opinion, the data on increased histone mRNA remain a weakness of manuscript. First, the data are somewhat disconnected from the rest of the manuscript. Second and more importantly, the functional connection of increased histone mRNA to quiescence as well as the mechanistic basis of the mRNA increase remain unclear.

Reviewer #5

(Remarks to the Author)

Thank you for the response to my questions. Upon reading the reviews and the responses, I have a few additional questions:

1. Reviewer 1 asked whether there is a change in the levels of 18S and 28S rRNA in cells with RPP40 depletion. This seems like a critical question: is there less rRNA available for the cells, or just a change in rRNA biogenesis without an effect on rRNA levels?

The authors responded that there is no change in relative abundance of 28S and 18S rRNA and state as data not shown that there was a modest reduction in total RNA from C40 cells on day 7.

Do the cells have less rRNA? Could the effects reflect a scarcity of rRNAs? Or is there an impact of reducing rRNA biogenesis even if there is an abundant supply of rRNAs.

2. A related question is whether RPP40 depletion reduces translation rate. Even though the authors show this data, it is difficult to interpret because it is not clear whether reduced translation reflects reduced availability of rRNA or tRNA, or if it is a consequence of the induced cell cycle arrest.

Version 2:

Reviewer comments:

Reviewer #1

(Remarks to the Author)

The authors have satisfactorily addressed the previous critique. In fact, they are to be commended for providing additional data using FISH-Flow and for changing the title to accurately reflect the mainstay.

Reviewer #2

(Remarks to the Author)

Reviewer #3

(Remarks to the Author)

Reviewer #5

(Remarks to the Author)

The authors have address my comments.

REVIEWER COMMENTS

Reviewer #1 (Remarks to the Author):

The authors use an array of cutting-edge approaches to demonstrate that targeted degradation of human RPP40, a protein subunit associated with RNase P and RNase MRP, triggers cellular quiescence. This central finding is of great interest to biology and medicine. Despite a large body of data and the work's significance, strengthening the study/manuscript will help bolster the mainstay. To lend cogency to the narrative, certain intriguing observations and seemingly inconsistent findings also warrant elaboration. During revisions, the authors should consider the following comments/questions.

We thank the Reviewer for the favorable assessment and constructive criticism of our study. Below, we address the Reviewer's comments point-by-point.

1. The statement in the Introduction that "there is little known about the mechanisms that mediate the transition between proliferation and quiescence" should be expanded appropriately. This nexus has been of longstanding interest (albeit not with a focus on RNase MRP) given the multifaceted ramifications of quiescence. For example, Adam et al. (1987) *Exp. Cell Res.* 169: 345-356 showed that quiescent mammalian fibroblasts could be stimulated to enter the cell cycle provided that an rRNA threshold was fulfilled. Moreover, there has been considerable effort by many investigators to understand the molecular switches essential for regulating wide-ranging processes during the activation of quiescent stem cells, and during developmental versus cellular quiescence. Thus, the Introduction would benefit from a brief consideration of the linkages between quiescence and stem cells/aging, and a more focused lens on specific knowledge gaps pertinent to this study.

We agree and have edited the Introduction section, as suggested. We included the cited reference (lines 85-87) and the following paragraph (lines 49-61):

"Much of the interest in understanding the quiescent state is due to its critical role in maintaining adult stem cells and tissue functions over the lifespan of the organism (PMID 31052375). Quiescent adult stem cells can reversibly exit the cell cycle to generate new differentiated cells, which are essential for tissue homeostasis and repair following injury. However, with aging or disease, the ability of adult stem cells to replenish missing or malfunctioning differentiated cells declines, contributing to the deterioration of tissue function (PMIDs 26785478, 26732838, 31248787). This decline has spurred significant research into the molecular switches that regulate stem cell activation, with the aim of developing strategies to enhance tissue regeneration and combat aging (PMIDs 26785478, 30602763). Although progress has been made in understanding how stem cells exit quiescence and re-enter the cell cycle, the mechanisms governing the transition from proliferation to quiescence remain less well understood—yet are equally crucial for both development and the long-term maintenance of tissue function (PMIDs 22084377, 33558315, 36923701).

2. The authors cite Goldfarb and Cech (2017) *Genes Dev.* 31: 59-71, who used the CRISPR/Cas system to generate RNase MRP-depleted strains (ref. 33). Goldfarb and Cech demonstrated that loss of the RNase MRP RNA results in proliferation arrest of HeLa cells. Since Goldfarb and Cech showed that heterogeneity in RNase MRP depletion was likely because some cells escaped either transfection or CRISPR cleavage, it is reasonable that authors of this study favored the inducible and reversible degron approach to deplete a protein subunit of RNase MRP. However, given that RPP40 is associated with RNase P and RNase MRP, the observed effects on quiescence cannot be assigned solely to depletion of RNase MRP, notwithstanding the comparison between depletion of RPP21 (N21) versus RPP40 (C40). The text should indicate clearly that a combined effect of

decreased RNase MRP and RNase P function cannot be ruled out (see #8 and #24 below). The pre-tRNA processing defect is more evident in C40 than N21 cells; the EdU assay to assess cellular proliferation shows that there is an effect even in N21 cells.

The reviewer is correct in noting that we cannot completely rule out a contributing effect of depleted RNase P to the observed phenotype. As suggested, we have revised the text to acknowledge this point (lines 161, 162), despite the phenotypic distinction between N21 and C40 cells, as well as the earlier observation by Goldfarb and Cech of proliferative arrest in RNase MRP-depleted cells (PMID 28115465). We also highlighted the effect of depleting RNase P alone on EdU incorporation, as suggested (lines 293-296).

3. Given the above caveat and the availability of many *Saccharomyces cerevisiae* RNase MRP mutants, including mutations in proteins that are associated with only RNase MRP and not RNase P, it seems that the quiescence trigger could have been unambiguously established in yeast. Was there a compelling reason to not leverage these mutant alleles?

There are important differences in pre-rRNA processing between yeast and mammals. Unlike in mammals, RNase MRP in yeast does not mediate the essential cleavage of pre-rRNA into segments destined for incorporation into the small or large ribosomal subunits, but plays a less critical role (PMIDs 7515008, 8846297, 28524231, 22342225, 16769905, 21849504, 8602511, 8247008). It is thus unlikely that results in yeast would transfer directly to mammalian cells.

We note that cells could also be induced into quiescence by inhibition of RNA polymerase I (Pol I), which is uniquely responsible for transcription of the 47S/45S precursor rRNA (see Fig. R13 below). This observation supports the conclusion that rapid inhibition of rRNA biogenesis can induce quiescence in mammalian cells.

4. Although RNase MRP RNA and protein levels are almost completely depleted in C40 cells, there remains some “non-canonical” processing of 47S/45S precursor rRNA. Is there another enzyme that is compensating for RNase MRP loss, albeit inefficiently/inaccurately? In addition, the pattern of non-canonical rRNA maturation products observed here is different from that observed when the MRP RNA subunit was depleted (ref. 33).

We do not find any evidence for mechanisms that would compensate for the loss of RNase MRP. In contrast, our northern blot analyses indicate that all prominent non-canonical products of pre-rRNA processing, which accumulate in RNase MRP-depleted cells, include an intact (uncut) ITS1 segment that is normally cut by RNase MRP. We now include an additional analysis with a probe complementary to the 5' ETS region of the pre-rRNA that validates this conclusion (Fig. R1A, included as revised Fig. 1I).

REDACTED

Figure R1. A (revised Fig. 1I) Northern analysis of rRNA precursors in C40 cells (n = 3) treated with DMSO or dTAG for 3 h or 24 h. Schematic on the left indicates where the northern probes 5' ETS (brown), mid (green), and 3' end

(purple), map to in pre-rRNA, as well as the structures of the detected pre-rRNA processing products (blue diagrams, canonical rRNA precursors; orange diagrams and asterisks, previously annotated noncanonical rRNA precursors; PMIDs 28115465, 23439679). Location of the annotated RNase MRP cleavage site within ITS1 is also shown. **B** Northern analysis of rRNA precursors from composite HeLa populations treated with control (c) or *RMRP* targeting CRISPR guides 1, 2, and 4 (copied from Fig. 4C in Goldfarb and Cech, Genes Dev 2017; PMID 28115465).

We note that our observed pattern of pre-rRNA processing is similar to that observed by Goldfarb and Cech (Fig. R1B, copied from PMID 28115465; compare to northern analyses in Fig. R1A that used the same probes, termed mid and 3' end). Notable differences include 1) stronger accumulation of unprocessed 47/45S pre-rRNA in our depletion experiment and 2) appearance at 24h of dTAG treatment of an additional non-canonical product with intact ITS1, which is consistent with the previously described product 30SL3' (PMID 23439679; Fig. R1A). We suggest that these differences result from a more rapid and thorough inhibition of rRNA biogenesis in C40 cells, enabled by inducible degradation of RNase MRP.

While it is useful to show these rRNA intermediates, what are the levels of 18S and 28S rRNA? Are mature rRNA levels altered at all? Please include these data.

We observe essentially no changes in the relative levels of 28S and 18S rRNA, in line with an equally important role of RNase MRP in maturation of 28S and 18S rRNAs (Fig. R2). We did, however, notice a modest reduction in the quantity of total RNA extracted from C40 cells after 7 days of treatment with dTAG compared to a similar number of DMSO-treated cells (not shown). We included the result shown in Fig. R2 as new Fig. S1K in the revised manuscript.

Figure R2 (new Fig. S1K). Relative levels of 28S and 18S rRNA in total RNA from DMSO- or dTAG-treated C40 cells analyzed by denaturing agarose gel electrophoresis.

5. Fig. 1D and Fig. 1E: The northern blot shows that RNase MRP RNA is nearly absent, yet the quantitation shows that both the 3-h and 24-h time-point have one-fourth or one-third (of the control) remaining. Are the relative expression levels also normalized based on the loading control?

To address this point, we quantified triplicate northern blot experiments, including the one shown in Fig. 1D, by normalizing the *RMRP* signal to *SNORD3A*, which served as a loading control. The results are comparable to those obtained via qPCR analysis shown in Fig. 1E, both suggesting that about 25% of the control *RMRP* level still remains in dTAG-treated cells (Fig. R3). We revised Fig. 1D to show a higher exposure of the *RMRP* northern blot.

Figure R3 (revised Fig. 1D). Northern analysis of *RPPH1* and *RMRP* from C40 cells ($n = 3$) treated with DMSO or dTAG for 3 h or 24 h. *SNORD3A* (U3 snoRNA) serves as a loading control.

6. The ectopic inducible expression of RPP40 to rescue the degron-mediated disruption of native RPP40 is an appealing approach. However, it is not clear why the kinetics of rescue are so different between the dTAG washout and the Dox-inducible system. Does the degron system display a delayed homeostatic reset (post-washout)? If this is the case, please cite appropriate literature.

The faster kinetics of the rescue by inducible RPP40 compared to the dTAG washout rescue is likely due to the known leakiness of the Dox-inducible systems. This is evidenced by weaker depletion of the catalytic RNAs and protein subunits of RNases P and MRP in dTAG-treated inducible C40 cells (Figs. 1D, E, G, 2D-F and S2D) that would be expected to promote more rapid exit from quiescence. Another contributing factor is the higher rate of Dox-driven transcription of ectopic *RPP40* compared to transcription of endogenous *RPP40*. The dTAG system in general allows for rapid recovery after washout (within 1 h; PMID 29581585).

7. The northern blot data for the dTAG+Dox condition must be included (i.e., primary data for Fig. S2C) because it is intriguing that RNase P and RNase MRP RNAs are recovering even before expression of FLAG-RPP40 (6-h timepoint). Or, does the variable limit of detection of these different measurements render a one-to-one correlation hard to establish? Please explain.

We now include a higher exposure of the FLAG-RPP40 immunoblot that shows substantial induction of the ectopic protein already at 6 h of treatment with Dox (Fig. R4; included as revised Fig. 2F). This can explain the observed early recovery of the catalytic RNAs detected by qPCR (not by northern blotting) now shown in Fig. S2D. This is also consistent with the likely rapid transcription of ectopic compared to endogenous *RPP40*.

Figure R4 (revised Fig. 2F). Immunoblot analysis of ectopic RPP40 (Flag-RPP40) in lysates of Dox-inducible C40 cells treated with DMSO or dTAG with or without Dox, as indicated ($n = 3$).

8. RNase P has been depleted using the degron system to dampen the levels of either RPP40 (C40) or RPP21 (N21), two different protein subunits of human RNase P. There are some inexplicable findings that merit mention and perhaps even some speculation as to the underlying bases. First, it is surprising that the levels of select mature tRNAs either stay the same or increase slightly (but not

decrease!) in C40 and N21 cells (Fig. 1H and Fig. S1J); the finding is particularly notable given the pronounced down-regulation of the RNase P RNA as well as several protein subunits in the C40 cells (compare Figs. 1D and S1F). It is possible that pre-tRNAs are subject to a higher turnover rate than their mature counterparts and the northern blot underestimates the 5' biogenesis defect. Worth including this perspective?

Mature tRNAs, like rRNAs, are highly stable (their half-lives in cells are on the order of several days; PMIDs 29463900, 20810645, 7400745) and their levels would not be expected to decrease at the times analyzed (3 h and 24 h in Figs. 1H and S1L). Given the high stability of mature tRNAs, the observed accumulation of pre-tRNAs, whose half-lives are only about 10 minutes or less (PMID 19287396), demonstrates strong inhibition of tRNA processing. Had we probed for mature rRNAs in dTAG-treated C40 cells, we would not have detected large accumulation of rRNA processing intermediates due to the high concentration of mature rRNA.

Second, the precursor-tRNA (pre-tRNA) accumulation appears to be identity dependent. For instance, tRNA^{Arg}(UCU) does not seem to be affected at all in C40 cells in stark contrast to N21 cells. Why?

We repeated the northern experiment for tRNA^{Arg}_{UCU} several times and consistently observed precursor accumulation comparable to that of other tRNAs in both C40 and N21 cells. We have revised Fig. 1H to show a representative tRNA^{Arg}_{UCU} northern blot (Fig. R5).

Figure R5 (revised Fig. 1H). Northern analysis of tRNA^{Arg}_{UCU} from C40 cells (n=3) treated with DMSO or dTAG for 3 h or 24 h. SNORD3A serves as a loading control. m, mature tRNA; p, precursor tRNA. Note that primary tRNA^{Arg}_{UCU} transcripts contain an intron that is removed separately from their 5' leader and 3' trailer pre-tRNA sequences, yielding an additional band by northern blot analysis.

Third, in the case of N21, only RPP21 is depleted. The other RNase P proteins (RPPs) seem largely unaffected. In C40 cells, however, the depletion of RPP40 results in loss of RPP21, RPP29, RPP38, RPP14, and RPP30. If the loss of H1 RNA is due to exposure of a naked RNA (uncoated with cognate protein subunits) to nucleases, one would have predicted a more severe reduction in C40 cells but this is not the case. So, what accounts for similar changes in H1 RNA levels in N21 and C40 cells? Are the relative levels of RNase P RNA versus RNase MRP RNA transcripts known? Does RNase MRP RNA somehow compensate for the loss of RNase P RNA and stabilize the free protein subunits?

Both RNase P and MRP are multisubunit enzymes. It is not surprising that, dependent on the subunit depleted, loss of other protein subunits will vary, particularly since some protein-protein subcomplexes may be stable. The important finding is potent inactivation of the targeted enzyme(s) in either case. Previous observations show that depletion of any of the RNase P/MRP protein subunits in bacteria, yeast, or human cells results in reduced levels of the associated catalytic RNA (PMIDs 19114042, 9620854, 23439679). An instructive finding is that in bacteria, the sole protein subunit of RNase P (C5), which is only one-tenth the size of the catalytic M1 RNA, can protect this relatively large RNA molecule from degradation (PMID 19114042). These observations argue that only the intact structures of RNases P and MRP protect the catalytic RNA from degradation, and that the level of protection does not correlate with the exposed area of the RNA or the number of depleted protein subunits. Although pools of common protein subunits are shared between the two enzymes, it is unlikely that *RMRP*, whose expression level is similar to the level of *RPPH1* in human cells, could stabilize close to twice its molar amount of protein subunits upon loss of *RPPH1*.

Fourth, POP5 and RPP14 each form a heterodimer with RPP30. It is not easy to appreciate why depletion of RPP40 would affect RPP14 and RPP30 but not POP5. Please explain/speculate.

Unlike other protein subunits of the palm module in human RNase P, including RPP14, RPP30, and RPP40, the POP5 subunit interacts with the finger module, thus connecting the finger and palm modules together (PMID 30454648 and Fig. 1A). We speculate that the protein-protein interaction of POP5 with the finger module stabilizes this subunit and protects it from degradation upon depletion of RPP40. This is consistent with the observation that all protein subunits of the finger module (RPP25, RPP20, POP1), and POP5 in the palm module, uniquely remain intact in RPP40-depleted cells (Fig. 1A, G). Given the extensive similarity between the overall architectures of RNases P and MRP in yeast (PMIDs 32586950, 32651392), it is likely that the stabilities of the common protein subunits of human RNases P and MRP, including POP5, are similarly affected upon rapid depletion of RPP40. We included this comment in the revised manuscript (lines 125-128).

Fifth, why is POP1 barely detected in C40 cells, even in DMSO controls, when it is readily detected at levels comparable to other protein subunits in N21 cells.

The immunoblots showing POP1 in C40 cells (Fig. 1G) or in N21 cells (Fig. S1I) were generated using two different lots of the anti-POP1 antibody (Abcam, ab254978). We revised Fig. 1G by replacing the initially shown POP1 immunoblot with one generated using the same antibody lot as used in Fig. S1I (lot #GR3288581-3), which yields stronger POP1 signal (Fig. R6).

Figure R6 (revised Fig. 1G). Immunoblot analysis of RNase P and MRP protein subunits in lysates of C40 cells treated with DMSO or dTAG for 24 h (n = 3).

Last, human RNase P RNA has multiple bands. Why? Fig. S1F also shows more than one species, but these appear to be smaller (unlike Fig. 1D).

The satellite bands seen in northern analyses of human *RPPH1* have been observed previously by others (e.g., Akiyama et al, bioRxiv 2022). Speculatively, these are *RPPH1* processing/degradation intermediates or transcripts of *RPPH1* pseudogenes.

9. Some of the RNA-seq data are puzzling. For instance, despite pre-tRNA build-up in both N21 and C40 cells, changes in the expression of tRNA processing-associated genes seem to be different (Fig. 4 and Fig. S3). What about tRNA transcription? Any data?

Our results demonstrate a coherent transcriptional induction of genes associated with translation and tRNA processing in C40 but not N21 cells, suggesting a countering response to blocked rRNA rather than tRNA biogenesis, as discussed in the manuscript (Figs. 4 and S4). This is substantiated by prior studies where induction of the same gene groups is observed upon specific inhibition of rRNA biogenesis, including the *RMRP* deletion study by Goldfarb and Cech (PMID 28115465; see also Fig. S4K, L) and the cited study of impaired rRNA processing in zebrafish (PMID 34019640). To address the question of the effects on tRNA transcription, we reanalyzed our TT_{chem} -seq data from dTAG-treated C40 and N21 cells. We found perturbed expression of several tRNA transcripts at 3 h and at 24 h of dTAG treatment in both cell lines, with more significant changes observed in C40 compared to N21 cells (Fig. R7). However, we found no overall bias towards increased or decreased expression of tRNAs, suggesting that the rRNA biogenesis-linked transcriptional induction is restricted to select subsets of protein-coding genes.

Figure R7. Changes in tRNA transcription in response to rapid depletion of RNases P and MRP. **A** Volcano plots showing differential tRNA abundances between C40 cells treated with DMSO or dTAG for 3 h (left) or 24 h (right; TT_{chem} -seq data; n = 3). Numbers above the volcano plots are counts of significantly regulated tRNA transcripts (adjusted p-value (padj) ≤ 0.01) in different bins separated by dashed vertical lines according to the strength and sense of regulation. Transcripts showing twofold or larger changes in abundance are highlighted in color. FC, fold change. **B** As in **A**, but showing the results for N21 cells.

10. Histone mRNAs are among the most highly upregulated transcripts upon depletion of RPP40. What about the levels of histone proteins (or even PTMs)? How does the increase in histone mRNAs correlate to cell cycle arrest? What happens to the levels of histone proteins upon entry into or exit from quiescence? The link is not clear. Moreover, why do the histone mRNAs form cytoplasmic punctae (Fig. 5C)? Are other RNAs also present in these bodies and are these punctae reversible liquid condensates? Based on the explanation provided for the 3'-processing of histone mRNAs, will incubation of the cytoplasmic extracts of dTAG-treated cells result in non-templated nucleotide

addition? Additional experimentation with histones is clearly necessary to build a complete picture. Thus, although the histone angle is interesting, the 3' processing aspect is a distraction from the RNase MRP mainstay. Clearly, direct versus indirect effects that ensue from downregulation of RNase MRP need to be parsed, but such an investigation is beyond the scope of this study. Best to focus on the theme that RNase MRP is one switch to toggle between quiescence and proliferation.

We agree that the persistence and upregulation of histone mRNAs is one of our most striking findings. We have included this data since it strongly supports the conclusion that a substantial fraction of the cells arrested in S phase and maintained S-phase gene expression. Note that both histone mRNAs and SLBP are only present in S-phase cells (PMID 18927579). As expected, overall histone gene transcription is reduced, but the histone mRNAs are stabilized, resulting in an increase in total histone mRNA levels over time (Figs. 4 and S4). There is no change in the levels of histone proteins in these cells. In any cell, at least 98% of the histone protein is associated with chromatin, and that protein is extremely stable (PMIDs 5364717, 17052464). If more histone protein is made, it is rapidly degraded.

The data on 3' ends shows the histone mRNAs are indeed stable, and their presence in cytoplasmic puncta suggest they are translationally silenced. Note that these are not stress granules, since they did not stain with any of a number of stress granule markers, as discussed in the manuscript (lines 315-317). Also note that the uridylation of histone mRNA 3' ends occurs after processing in the cytoplasm, not as part of the processing.

We feel these data are important, since typically when cells arrest in S phase due to inhibition of DNA replication, histone mRNAs are rapidly degraded, pointing to a difference compared to the arrest that occurs in the dTAG-treated cells.

Other comments

11. Text and figure edit (global): 5' not 5'; 3' not 3'.

We corrected this.

12. The Introduction ends rather abruptly. This section would benefit from a final short paragraph summarizing the goal(s) of the study, the key findings, and the main takeaways.

We included a short summary at the end of the introduction, as suggested (lines 97-101).

13. TTchem-seq should be defined and briefly described as readers are less likely to be familiar with this relatively recently developed method compared to traditional RNA-seq.

We included a brief description of the TT_{chem}-seq method in Fig. S4F in the revised manuscript (Fig. R8).

Figure R8 (new Fig. S4F). Overview of the transient transcriptome sequencing with chemical RNA fragmentation (TT_{chem}-seq) method for profiling nascent transcription (adapted from Gregersen et al, Nat Protoc 2020; PMID: 31915390). Nascent RNA is pulse-labeled by incubating cells in culture with 4-thiouridine (4SU) for 10 min. Total RNA is then extracted and fragmented by controlled base hydrolysis. The 4SU residues in the fragmented RNA (red)

are biotinylated using a biotin linker and captured with streptavidin beads before strand-specific library preparation and high-throughput sequencing. RNA pol, RNA polymerase.

14. Fig. 1A: Protein subunits not drawn to scale (e.g., compare POP5 versus RPP14). Why?
We drew the schematic of human RNase P based on the surface representation shown in Fig. 1C of the cryoEM study by Ming Lei and coworkers (PMID 30454648).

15. Fig 1F: Quite useful. Why is an equivalent not provided in Fig. S1?
We now include, as new Fig. S1H, an RNA FISH analysis of *RPPH1* and *RMRP* in dTAG-treated N21 cells, akin to the analysis of C40 cells shown in Fig. 1F (Fig. R9).

Figure R9 (new Fig. S1H). RNA FISH of *RPPH1* and *RMRP* in N21 cells treated with DMSO or dTAG for 3 h. Scale bar, 20 μ m.

16. Fig. 1G (and elsewhere): Complete western blots (i.e., corresponding to the entire gel that was probed) should be included in the supplement. At least one representative blot for each protein tested. This comment is motivated by the fact that the western blot for RPP29 shows two bands and only one of these bands is affected post-dTAG treatment (Fig. 1G). In fact, the situation is even less clear in Fig. 2D where there is cyclical change in the intensity of one of these two bands. Likewise, Fig. 2D shows two bands for RPP20. Significance?

We now show the uncropped western blots used for the traces in Figs. 1G and 2D, along with other unprocessed data, in the Source Data file. The faint band that appears at a higher-than-predicted molecular weight (MW) in the RPP20 blot in Fig. 2D is likely non-specific, while both bands in the RPP29 blots in Figs. 1G and 2F are likely specific, as their intensities change coherently.

17. Fig. 1G, line 109: While an RNase P/MRP aficionado might understand this comment on finger-to-wrist assembly pathway, it would be helpful to include a better explanation based on the findings of Engelke, Krasilnikov, and co-workers.

We replaced this comment with a reference to the general topology of human RNase P and the similarity between the overall architectures of RNases P and MRP in yeast (lines 129-132; PMIDs 30454648, 32586950, 32651392).

18. Fig 1H and Fig S1J: The tRNAs should read GUG (not GTG), GUU (not GTT), and UCU (not TCT).

This has been corrected.

19. Fig. 1I: Labels do not appear to be consistent with the annotated bands. What does the orange-colored asterisk indicate? What is meant by “orange diagrams” in the legend? Binding sites of ‘mid’ and ‘3’ end’ probes are not clear from this figure. The solid line-format in Goldfarb and Cech (2017) may be easier for the reader.

We clarified these points in the revised Fig. 1I (see also Fig. R1A).

20. Fig. 2A: Please change the color of the line indicating “cell viability” as it likely to be confused with some other similarly-colored lines in the same panel.

We now use a distinct color (green) for the line indicating cell viability, as suggested.

21. Fig. 2C: The y-axis labels are not easy to comprehend. Please revise. Also, it is not clear why cells that have entered quiescence would respond differently in terms of the day of washout?

We have revised the y-axis labels in Fig. 2C to ‘Days to first increase in cell #’ and ‘Days to 10x increase in cell #’ to improve clarity. The timing of dTAG washout directly influences the duration of quiescence. As shown in Fig. 2C, cells that remain quiescent for longer periods require more time to re-enter the cell cycle upon washout. This observation is consistent with findings from several previous studies of cellular quiescence (PMIDs 2654144, 2766357, 29417058, 31636214, 30735649).

22. Fig. 2D and Line 142: What is meant by “bulk of each enzyme”?

By "bulk of each enzyme," we mean the predominant proportion of the examined protein subunits and the RNAs that constitute RNases P and MRP.

23. Fig. 2H: The range of the x-axis in the dTAG (14 day) panel is narrower than in all of the other panels. This trend is not replicated to the same extent in the N21 cells (Fig. S2F). Rationale?

The last sampled time-point was analyzed by flow cytometry separately from all other samples and thus shows different absolute intensity values for DAPI and EdU-AF594. Importantly, however, this sample is appropriately compared to its matching DMSO-treated control sample, as are all analyzed samples of C40 (Fig. 2H) and N21 cells (Fig. S2G). This ensures consistency in the analysis across all conditions.

24. Fig. 2I and Fig. S2G: What is the interpretation here? Do these data not support the contention that there is a combined effect of RNase P and RNase MRP while examining the EdU data?

We indeed cannot rule out a contributing effect of depleted RNase P; please see our response to point 2.

25. Fig. 2J and line 298: How does fig 2J show that the distribution of C40 cells in S phase is either 30% normally cycling or 35% quiescent? Unclear.

We replaced 'reside in S phase' with 'contain a DNA amount typical of S-phase cells' to clarify this point (line 351).

26. Fig. S1J: Is the additional band for pre-tRNA^{Arg} because of an intron-containing intermediate?

Yes, primary *tRNA^{Arg}_{UCU}* transcripts contain an intron that is removed separately from their 5' leader and 3' trailer pre-tRNA sequences, yielding an additional band by northern blot analysis. We now clarify this in the legends to Figs. 1H and S1L in the revised manuscript.

27. Fig. 3: Many upregulated transcripts in C40 correspond to genes involved in tRNA processing and translation. How does the transcription rate of tRNAs change?

Please see our response to point 9.

28. Fig. 4C: Color ramp key?

We have included the color gradient key in the revised manuscript.

29. Fig. 5E: Lacks the final two nucleotides (UA) in Fig. 5F. Fix.

30. Fig. 5F: Explain the colors and bar graphs. Unclear what is the message with respect to uridylation.

Points 29 and 30: Formation of mature histone mRNA 3' ends requires endonucleolytic cleavage in the nucleus, which leaves a 5-nt tail (ACCUA in the case of *H2AC13*), followed by trimming in the cytoplasm to leave a tail of 3 nts (ACC; Fig. R10). If the tail is shortened to <3 nts, uridines are added to maintain the length of the tail at 3 nts (lines 331-334 and PMID 28867047). Fig. 5E shows the most common 3' tails (ACC, ACU, AUU) of mature replication-dependent histone mRNAs. Fig. 5F shows EnD-seq data for *H2AC13* with data (bars) at positions of the last templated nucleotides; colors indicate the diversity of lengths of nontemplated tails (>99% of nucleotides in the nontemplated tails are uridines). The EnD-seq data are shown for the precursor mRNA with a 5-nt tail, offering a detailed view of the 3' termini, including insights into trimming efficiency in the cytoplasm. Data are shown for dTAG-treated C40 cells (3 h and 24 h) and DMSO-treated controls. Similar visualizations of EnD-seq data were used in previous studies (PMIDs 26015596, 30408609). We included Fig. R10 as new Fig. S5B in the revised manuscript.

Figure R10 (new Fig. S5B). Formation of a mature histone mRNA 3' end requires endonucleolytic cleavage in the nucleus, which leaves a 5-nt tail after the stemloop, followed by trimming in the cytoplasm to leave a tail of 3 nts (ACC). If the tail is shortened to <3 nts, uridines are added by the terminal uridylyl transferase TUT7 to maintain the length of the tail at 3 nts.

31. Fig. 6: The arrow and right-hand part are moot. Delete. (The figure itself does not add value.)

32. Line 78: Please include a citation.

33. Line 174: Replace “downturn” with “inhibition” or another suitable word.

34. Line 246: What is meant by “highly induced”

35. Line 252: Please include a citation.

36. Line 274: Are the figures cited here correct?

37. Line 276 (and elsewhere): Southern is a proper noun (thus, Southern blot), but northern is not.

38. Methods: (a) Please use SI units for time (h, min, s). (b) Specify % in either v/v or w/v.

39. References: Titles need to be uniformly formatted in sentence case (not title case).

40. Welting et al. (2006) RNA 12: 1373-1382 should be included in the list of citations, as this paper showed the association of RPP40 with RNase MRP.

Points 31–40: We addressed these points, as suggested. We replaced "highly induced" with "increased by more than 2-fold" (lines 341).

Reviewer #2 (Remarks to the Author):

Reviewer #3 (Remarks to the Author):

Reviewer #4 (Remarks to the Author):

MRP impacts on rRNA biogenesis and induces a quiescence cell cycle state. This is accompanied by acute transcriptional reprogramming and a decline in critical cellular functions. Interestingly, the authors also observed a strong increase in the levels of histone mRNAs, which accumulate locally in the cytoplasm.

Altogether, the findings are of interest, but overall remain at a rather descriptive level. In particular, it remains unclear whether and how the induction of histone mRNAs is connected to the quiescence phenotype. Further, it is long established that impaired ribosome biogenesis impacts on cell-cycle progression by activating a p53-dependent impaired ribosome biogenesis checkpoint (IRBC). The authors need to dissect to what extent the quiescence phenotype is connected to the canonical IRBC and why cells enter G0 phase. In this context they need to compare p53-deficient and proficient cells and additionally should explore the contribution of the pRB-axis.

We thank the Reviewer for the encouraging comments and constructive suggestions, which we address in the paragraphs below.

We agree that the p53-dependent IRBC, mediated by the ribosomal protein (RP)–HMD2–p53 signaling pathway (PMID 27908926), could be responsible for the observed cell cycle arrest and the induction of quiescence in C40 cells. In fact, this would be in line with previous studies in which perturbation of rRNA processing was found to engage the RP-HMD2-p53 pathway, though the reported cellular phenotypes were often not thoroughly examined or causatively linked to this pathway (PMIDs 18469340, 19087264, 34319761, 27908926).

To investigate the role of p53-dependent IRBC in quiescence induced by rapid inhibition of rRNA processing, we stably silenced the expression of p53 in C40 cells with a potent p53-targeting shRNA described previously (PMID 18614011; Fig. R11). We monitored the proliferation rate and viability of the p53-deficient cells (C40^{shP53}) alongside control C40 cells expressing a non-targeting shRNA (C40^{shCTRL}) during treatment with DMSO or dTAG and after washout. Despite undetectable p53 protein levels and efficient inducible depletion of RPP40 (Fig. R11), dTAG treatment led to a complete proliferative arrest of C40^{shP53} cells by day 3 (Fig. R11). Upon removal of dTAG at day 5, the arrested cells resumed proliferation after about four days of washout (Fig. R11B). Compared to control cells, C40^{shP53} cells exhibited slightly delayed arrest and moderately higher proliferation rate upon cell cycle reentry, while cell viability remained high (>80%) in both cell populations at all sampled times (Fig. R11B). These results suggest that cellular quiescence induced by rapid depletion of RNase MRP occurs largely independent of p53-mediated IRBC.

Figure R11 (new Fig. S3A and S3B). p53-independent induction of cellular quiescence by rapid depletion of RNase MRP. **A** Immunoblot analysis of p53 and HA-tagged endogenous RPP40 in C40 cells stably expressing p53-targeting shRNA (C40^{shP53}) or non-targeting control shRNA (C40^{shCTRL}) treated with DMSO or dTAG for 24 h. Actin

serves as a loading control (n = 3). Note that there was no obvious increase in p53 as a result of RPP40 depletion. **B** Growth of C40^{shCTRL} (gray) and C40^{shP53} (red) cells over the indicated periods of time. Cells were treated either continuously with DMSO (thin lines) or with dTAG for 5 days and then dTAG was washed out (WO; thick lines). Dotted lines indicate cell viability. Data are shown as mean ± SD (n = 3). *, p < 0.05; **, p < 0.01; ns, non-significant (Student's t test comparing C40^{shP53} dTAG and C40^{shCTRL} dTAG data).

To gain further insight into the observed proliferative arrest, we used immunoblotting to survey a number of key factors associated with p53-dependent or independent response to impaired ribosome biogenesis (p53, p21, HDM2, (phospho)Rb, E2F; PMID 25482194) and/or regulation of cell cycle (cyclin A, PCNA, CDC6; PMID 36923701). We found that p53, whose protein levels increased only about 2-fold over 7 days of dTAG treatment, was largely responsible for the induction of its downstream effector p21 (CDKN1A) at longer times of dTAG treatment when C40 cells begin to enter the induced quiescence (Fig. R12 and Figs. 2, 3). We observed little change in the levels of other examined factors, including (phospho)RB or E2F (Fig. R12). It is therefore possible that the mechanism may involve the observed global translational decline, which faithfully tracks with the decreasing rate of EdU incorporation and proliferative arrest (Figs. 2H and 3). We have included these results and their interpretation in the revised manuscript (Fig. S3 and lines 197-221).

Figure R12 (new Fig. S3C). Immunoblot analysis of factors associated with p53-dependent or independent response to impaired ribosome biogenesis and/or regulation of cell cycle. C40^{shCTRL} or C40^{shP53} cells were analyzed at the indicated times of treatment with dTAG.

While it is unclear how histone mRNAs might participate in the induced quiescence, we note that the persistence and upregulation of histone mRNAs in cells induced to quiescence is in itself a striking new finding, and we decipher its molecular basis in some detail. In particular, while histone mRNAs are not expected to accumulate in arresting cells that are not synthesizing DNA, our results suggest that translational decline along with low levels of their continued transcription in arresting cells create unique conditions that lead to stabilization of normally processed histone mRNAs. This provides a foundation for their further studies, including testing of our speculation that the accumulated histone mRNAs may facilitate reentry of quiescent cells into the cell cycle (lines 399-401).

Furthermore, the authors need to validate whether their observation can be expanded to depletion of other regulators of rRNA biogenesis. Use of an RNA polymerase I inhibitor and depletion of a set of other rRNA processing factors would be needed for generalizing their observation. Without providing more mechanistic insight and generalizing their observation, the findings are of limited significance.

Given that RNA polymerase I (Pol I) specializes in transcription of 47S pre-rRNA, which is the first event in rRNA biogenesis, Pol I inhibition might indeed be expected to effectively mimic the blockage in ribosome biogenesis achieved via inhibition of upstream rRNA processing, including the pre-rRNA cleavage mediated by RNase MRP.

We tested whether the pharmacological inhibition of Pol I would phenocopy the quiescence induced by rapid depletion of RNase MRP. Treatment of HEK293T cells with CX-5461, a highly potent first-in-class Pol I inhibitor (PMID 21159662), caused a complete proliferative arrest by day 2, with DNA synthesis essentially ceasing by day 3 (Fig. R13). The arresting cells showed distinct accumulation in G2 phase, in line with previous studies in different cell types (PMIDs 27729807, 27391441, 26061708), and remained viable for the duration of the experiment (Fig. R13). Strikingly, removal of the inhibitor on day 4 allowed the cells to resume proliferation within the following 4 to 5 days, as would be expected from C40 cells after 4 days of treatment with dTAG (Fig. R13A and Fig. 2A). Both dTAG-mediated degradation of RNase MRP and inhibition of Pol I by CX-5461 caused virtually all cells to exit cell cycle and stay alive, but with the stage of the cell cycle exit - either throughout the cell cycle or G2 - depending on the inducing stimulus.

These results demonstrate that quiescence can be induced by rapid inhibition of rRNA biogenesis that is not strictly dependent on the method of inhibition (genetic versus pharmacologic) or the targeted step in rRNA biogenesis (transcription versus early processing). We speculate that, akin to inhibiting the pre-rRNA cleavage by RNase MRP, blocking of other essential rRNA processing steps may similarly induce cellular quiescence. Although we have not explored this point systematically, we note that previously reported transient inhibition of essential rRNA processing factors, including LAS1L, PELP1, and NOP2, as well as inhibition of Pol I, resulted in growth arrest but sustained cell viability, consistent with this possibility (PMIDs 34319761, 31727736).

Figure R13. Inhibition of rRNA transcription induces cellular quiescence. **A** Growth of HEK293T cells treated either continuously with phosphate buffer (Control; gray), with the Pol I inhibitor CX-5461 at 2 μ M for 4 days and then washed out (WO; light green), or continuously with CX-5461 (dark green, (on)). Dotted lines indicate cell viability. Data are shown as mean \pm SD (n = 3). **B** Cell cycle analysis of CX-5461 treated HEK293T cells by flow cytometry.

Cells were treated with CX-5461 at 2 μ M for the indicated times, pulse-labeled with EdU for 1 h and stained for the incorporated EdU (with Alexa Fluor 647, AF647) and DNA (with FxCycle Violet Stain), then analyzed by flow cytometry. **C** Phases of the cell cycle quantified as indicated by the gating in **B**.

Reviewer #5 (Remarks to the Author):

In this manuscript, Liu and colleagues investigate the effects of depleting RNase MRP in human cells. MRP splits pre-rRNA into segments that will be incorporated into small or large ribosomal subunits. After this enzyme is depleted, there is a decrease in the synthesis of ribosomal RNAs. The cells stop cycling and accumulate histone messenger RNAs. The cells stop cycling at different stages of the cell cycle and can remain in this state for weeks and alive. Restoring expression of MRP results in resumed proliferation at the speed of control cells.

Comments

Overall

The rapid depletion of human RNases P and MRP with the conditional degron approach provides for a first understanding of the impact of removing these two proteins. The ability to wash out the degradation inducing dTAGs represents an advantage for these studies because it is possible to test the reversibility of these effects.

We agree with the reviewer. Our study is the first to rapidly deplete RNase MRP and demonstrate a reversible arrest of cellular proliferation, which we have characterized in some detail. We strongly feel that this is a substantial contribution that stands on its own, as an initial paper in this field.

If the authors would like to argue that depleting RNase P and MRP induces cells into quiescence, it would be better to perform these studies in cells that enter a quiescent state in physiological conditions, for instance, primary cultures and stem cell models. The 293 cells used here are so far removed from being in a body and likely have so many cell cycle changes that they are not a good choice for studies of quiescence. HCT116 is also a poor choice because cancer cells do not become quiescent the way that primary human cells do.

To distinguish quiescence induced via molecular targeting and naturally occurring quiescence in multicellular organisms, we now refer to the former as 'induced quiescence' and have revised the manuscript accordingly.

Physiological signals that induce mammalian cells into quiescence are poorly understood and are thought to be highly complex, consisting primarily of altered local availability of different growth factors and cytokines, which act in a cell type- and niche-specific manner (PMIDs 30735649, 33171109). As such, physiological stimuli of quiescence induction are not readily amenable to *in vitro* studies. In fact, unlike studies of the maintenance or exit from quiescence, induction of quiescence in mammalian cells has primarily been achieved via non-physiological means such as serum withdrawal or loss of adhesion (PMID 16509772). Moreover, primary human cells can only undergo a limited number of divisions in culture, prohibiting studies that involve complex genome editing, derivation of monoclonal lines, and long-term experimentation employed in our work.

Importantly, our intent was not to study a specific cellular setting where a specialized trigger would induce quiescence via signaling, but rather to understand molecular mechanisms that underlie quiescence as a conserved cellular phenotype. Thus, we consider the capacity to induce quiescence

in broadly used cell lines as diverse as the 'normal' HEK293T and cancerous HCT116 cells, neither of which has, to our knowledge, been induced to quiescence before, a powerful demonstration of the potency and a candidate generality of the identified quiescence-inducing trigger.

Use of a specific (physiological) quiescence-inducing signal to prove that reduction in rRNA biogenesis is causative of the phenotype would make the task essentially impossible. In fact, the myriad of specific signaling-associated effects – among them on rRNA biogenesis – has historically confounded identification of core principles that may be common to cellular quiescence (PMIDs 22084377, 33171109, 23698583, 16509772, 19833516).

For the above reasons, and with the caveat that not more cell types could be analyzed in this initial report, we would like to argue for the appropriateness of the cellular systems and the genetic technologies employed here. Based on their critiques, we also believe that the other Reviewers agree with our choice of the cellular systems.

A cell that is not proliferating isn't necessarily quiescent. If the authors argue that loss of these RnaseP/MRP is sufficient to make cells quiescent, the authors should characterize the state they have entered to see if signature hallmarks of quiescence are present. Johnson and Cook lay out characteristics of quiescent cells (Fac Rev 2023 12: 5). These are listed in Table 1 as reduced cyclin expression, increased expression of CDK inhibitors, low CDK activity, unlicensed DNA replication origins, low RB phosphorylation, and others.

We agree and do not claim that any non-proliferating cell is quiescent. Cellular quiescence is defined as a state of reversible proliferative arrest in which cells are not actively dividing but retain the capacity to reenter the cell cycle (PMIDs 31189093, 33171109). Our findings demonstrate that the cellular state induced by rapid depletion of RNases P and MRP faithfully conforms to this definition of cellular quiescence. We establish this not only by monitoring cellular proliferation and viability, but also by documenting the quiescence-associated decline in critical cellular functions, including cessation of DNA synthesis and dramatically reduced rates of translation, metabolism, and gene transcription, which are considered the most reliable and general signature features of cellular quiescence (PMIDs 30976106, 35417196, 22084377).

There is considerably less consensus on specific molecular markers of quiescence and, in fact, no one marker or a set thereof exists that unequivocally identify quiescent cells, reflecting the large diversity of regulatory schemes of cellular quiescence (PMIDs 35417196, 22084377, 33171109, 23698583, 36449357). Commonly used 'markers' of quiescence are almost exclusively ubiquitous cell cycle regulators, including cyclins, CDK inhibitors, the retinoblastoma protein (RB), its CDK-mediated phosphorylation, and its regulation of the E2F class of transcription factors, the p53 tumor suppressor, regulators of DNA replication, and others (PMIDs 31189093, 36923701). No known marker exists that would predict reversibility of proliferative arrest, a key defining feature of quiescence. Note also that the large majority of the common marker proteins regulate progression of cells through G1 phase where most studied cases of quiescence initiate, but this is of limited relevance to forms of quiescence that is entered from other cell cycle phases, which we also observe in our study (PMIDs 8467507, 21402786, 12631573).

To further characterize the quiescence induced by rapid depletion of RNases P and MRP, we immunoblotted several proteins, as recommended by the Reviewer, that are commonly studied in the context of cellular quiescence (PMIDs 31189093, 36923701; see also Fig. R12). We detected increased expression of the p53 tumor suppressor as well as induction of its regulated CDK inhibitor p21 (CKDN1A) at longer times of dTAG treatment, although these changes are not necessary for induction of quiescence, as we show above (Fig. R12 and Figs. 2, 3). We observed little change in the levels of several other examined factors, including (phospho)RB or E2F1, in line with the

quiescence entry from multiple stages of the cell cycle. We have included these results and their interpretation in the revised manuscript (Fig. S3 and lines 197-221).

The authors should choose a model system in which the cells can become quiescent in vivo and test the impact of depleting Rnase P and MRP in this system. They can also perform RNA seq on RnaseP and MRP depleting cells, proliferating cells, and cells induced into quiescence by different signals and determine whether the RNase P/MRP depletion results in gene expression changes that are similar to the relevant quiescence signals for the chosen cell type.

Please see above our biological and technical arguments against experiments using different inducing signals as well as arguments for the appropriateness of the cellular systems employed in our study.

In their abstract, the authors argue that targeting rRNA biogenesis is a general mechanism for the induction of cellular quiescence. To make this claim, the authors should show that in at least their chosen quiescence model, and for this claim, multiple models would be needed, that physiologically relevant quiescence signals result in decreased rRNA biogenesis and changes similar to that observed in their MRP degradation experiments. Ideally, the authors would then show that if they apply a relevant quiescence signal, but prevent the cells from downregulating rRNA biogenesis, that the cells fail to enter a proper quiescent state.

Decreased rRNA biogenesis is observed broadly in physiologically occurring quiescence, from bacteria to human cells, and restricting the supply of rRNA can cause cell cycle arrest with sustained viability (PMIDs 3556422, 35417196, 8203157, 15568992, 6317207, 9432720, 10542411, 34319761, 31727736). Notably, Ki-67, one of the most commonly used 'markers' to distinguish proliferating from quiescent cells, is itself a protein involved in rRNA biosynthesis (PMIDs 16206250, 17531085, 28497998).

Fortifying the argument that targeting rRNA biogenesis presents a mechanism for induction of cellular quiescence, we find that pharmacological inhibition of RNA polymerase I (Pol I) similarly induces cellular quiescence, with kinetics of proliferative arrest and reentry comparable to those upon rapid depletion of RNase MRP (see Fig. R13). Thus, inhibition of rRNA biogenesis can induce quiescence in a manner that is not strictly dependent on the method of inhibition (genetic versus pharmacologic) or the targeted step in rRNA biogenesis (transcription versus early processing). Note also our use of the term 'induced quiescence' to refer to quiescence induced by rapid inhibition of rRNA biogenesis.

Specific comments

1. The authors state that all quiescent cells have remarkably low protein biosynthesis rates but only cite a review in yeast. Such a statement would need more validation across many quiescent systems.

We agree and now cite several studies that document low protein biosynthesis rates during quiescence across different cell types and species (PMIDs 8083172, 24670665, 27492367, 26549106, 28235822, 30038306, 32858411, 1383195; lines 71-73).

2. The authors say that it remains to be explored whether reducing rRNA biogenesis. There is a substantial literature on rapamycin which reduces rRNA biogenesis and has been studied extensively as a mechanism for inducing a cell cycle arrested state.

Rapamycin is a specific inhibitor of the TOR protein kinase that, among other, regulates cell growth and proliferation, cell survival, and protein synthesis. One of the multiple effects of rapamycin is indeed a reduced rate of rRNA biogenesis; however, the net effect of rapamycin cannot be ascribed specifically to the inhibition of rRNA biogenesis. We included in the discussion a comment on the inhibitory activity of rapamycin and its capacity to induce a quiescence-like state in pluripotent cells (PMIDs 27880763, 22121221; lines 379, 380).

3. In Fig 1, the authors show an effect of the ablation on tRNA and rRNA. What happens to mRNAs?

The nascent and steady-state levels of mRNAs are analyzed and discussed in detail in Figs. 4, 5, S4, and S5. Please also refer to the corresponding sections of the text.

4. Fig 2B. Those cells look sickly, possibly like they are apoptosing. A viability dye may not pick up early apoptosis. Quiescence is not usually associated with a visible rounding and detachment from the plate.

The dTAG-treated C40 cells, such as those shown in Fig. 2B, remain >85% viable for several weeks, as documented at numerous time points by staining with Trypan Blue, analysis of DNA content by flow cytometry, microscopy, and their capacity to resume proliferation upon washout. Early apoptosis at any time point would lead to cell death at a later time, which we do not observe. However, to eliminate any uncertainty, we quantified the proportions of viable, dead, and apoptotic cells at different times of dTAG treatment using the Apoptosis Kit (ThermoFisher, Cat. nr. A35136; Fig. R14). This assay confirmed high viability of arresting and quiescent cells, in line with the above analyses. We included the result shown in Fig. R14 as Fig. S2C in the revised manuscript.

Figure R14 (new Fig. S2C). Viability of dTAG-treated C40 cells analyzed by the Apoptosis Kit (ThermoFisher, Cat. nr. A35136). Cells were treated with dTAG for the indicated periods of time or with puromycin at 2 $\mu\text{g}/\text{ml}$ for 12 h (Puro), co-stained with annexin V to detect externalized phosphatidylserine in apoptotic cells and SYTOXTM AADvancedTM stain to detect dead cells, and analyzed by flow cytometry. Proportions of viable, apoptotic, and dead cells were determined according to the manufacturer's instructions.

We note that yeast and mammalian cell can exhibit visible rounding when they enter quiescence (PMIDs 36794724, 27738016, 8799823, 23698583) and weaker attachment to the culture plate compared to proliferating cells (PMIDs 23839578, 8106557, 29803144, 688399). This behavior aligns with our observations of quiescent cells.

5. In Fig 2H and J, the authors show that the cells stop in every cell cycle phase upon depletion of RNaseP/MRP. It is true that some examples of cells entering quiescence in phases other than early G1 have been discovered. But it is still typical for cells to enter quiescence in early G1. The authors should perform these experiments in cells that enter quiescence in response to physiological signals, treat those cells with quiescence inducing signals or inactivate RNaseP/MRP and determine whether

the cell cycle phase distribution produced is similar or different. If quiescence signals induce the cells to exit in G1 and RNaseP/MRP inactivation results in a range of cell cycle states, this would not support the authors' conclusion that rRNA biogenesis inactivation is the common trigger for quiescence.

Our results, supported by several prior observations of arrested cell proliferation with sustained viability upon blockage of rRNA biogenesis (PMIDs 34319761, 31727736, 37356716, 9872334, 27880763, 26871632), strongly suggest that inhibition of rRNA biogenesis below a certain threshold can induce cellular quiescence. The rate of rRNA biogenesis naturally drops to its lowest levels during the early G1 phase, following mitotic shutdown, making this phase particularly prone to quiescence entry (PMID 10339547). Indeed, previous studies suggest that the preferential quiescence entry in G1 is a consequence of metabolic slowdown rather than necessity (PMIDs 21402786, 21554667, 12631573, 8467507, 28497998). It is thus conceivable that in the reported examples of cells entering quiescence in different phases of the cell cycle, akin to the induced quiescence in our system, the rate of rRNA biogenesis drops sufficiently low in G1 as well as in other phases.

It is reasonable to assume that direct and rapid inhibition of rRNA transcription or processing reduces the rate of rRNA biogenesis to a lower level and across a broader portion of the cell cycle than most other quiescence-inducing signals. Therefore, it would not be unexpected for a specific signal to drive cells into quiescence in G1, while inhibition of rRNA biogenesis leads the same cell population to enter quiescence at a different phase of the cell cycle.

6. The authors observe a large induction of transcripts for ribosome biogenesis, translation initiation and elongation, regulators of tRNA processing, and histone mRNAs when RNaseP/MRP are inactivated. If ribosomal biogenesis inactivation is a common mechanism of quiescence induction, are these same gene categories induced when cells enter quiescence?

There is no reason to think this would be the case. The cells are likely responding to the acute loss of rRNA biogenesis by transiently upregulating the large majority of the factors in these pathways. This is strikingly similar to the 'countering' responses in microorganisms to quiescence-triggering lack of specific nutrients or the transcriptional response to impaired rRNA processing in zebrafish, as we discuss in the manuscript (PMIDs 29432178, 36794724, 34019640).

7. What is the authors' model for why histone mRNAs accumulate with RNase P/MRP knockdown? Why are histone mRNAs more stable but other transcripts expressed in S phase are not?

Histone mRNAs are the only protein-coding transcripts without a polyA tail and their normal processing is unique in several aspects. One is that their degradation requires SLBP and active translation. We show that SLBP continues to be expressed at normal levels, while translation is dramatically reduced in cells entering quiescence. Our results suggest that the increased stability of histone mRNAs resulting from translational decline together with their continued transcription leads to accumulation of histone but not most other mRNAs in the induced quiescent cells.

8. In the last sentence, the authors speculate that in early G1, rRNA biosynthesis is naturally low, so that is why many cells exit the proliferative cell cycle in G1. Is there any evidence to support this argument? There are cells that are reported to enter quiescence from other cell cycle phases. Is there any evidence that they have low ribosomal biogenesis at an unusual time in the cell cycle that's driving this? Can the authors test this by increasing ribosomal biogenesis and determine whether cells that normally quiesce now do so at a different phase of the cell cycle?

Please see our responses to the general comments and point 5. It is well established that, other than in mitosis, the rate of rRNA biogenesis is at its lowest in early G1 (PMIDs 13775619, 14488623, 10339547, 9857193). It has, to our knowledge, not been examined whether rRNA biogenesis is similarly decreased in other phases of the cell cycle in systems where quiescence is entered from a different phase. However, given that a decreased rate of rRNA biosynthesis is a general trait of quiescence, observing low rates at any particular stage cannot distinguish cause from effect. We also do not believe that increasing ribogenesis, as suggested, without affecting several other cellular processes is experimentally feasible. Proving our speculation that naturally low levels of rRNA biogenesis in early G1 favor quiescence entry from this stage will therefore require considerable additional work, but this is beyond the scope of this study.

We thank the reviewers for their constructive comments that have allowed us to greatly improve the manuscript through two rounds of revision. Below, we provide a point-by-point response to the remaining comments from each reviewer. Additionally, we addressed an editorial request—aligned with a reviewer’s suggestion—to describe our observations as ‘cell cycle arrest’ rather than ‘induced quiescence’ in the title and main text. We now use the term ‘reversible proliferative arrest’ and discuss quiescence as a phenotype of primary cells observed in nature. Accordingly, we have revised the manuscript title to “Reversible proliferative arrest induced by rapid depletion of RNase MRP”.

REVIEWER COMMENTS

Reviewer #1 (Remarks to the Author):

The authors have strengthened the manuscript with various revisions. As stated in a review of the previous version, the authors’ central finding that downregulation of RPP40, a protein subunit associated with RNase P and RNase MRP, triggers cellular quiescence is of great interest to biology and medicine. Thus, the work warrants publication. However, the authors are encouraged to consider the following comments/questions during the next (final?) round of revisions.

1. In response to a previous critique that it would be desirable to document the levels of 18S and 28S rRNAs post-depletion of RPP40, the authors now show these data in Supplementary Figure 1K. Moreover, they comment in the text that the levels of these rRNA do not change upon RPP40 depletion. Given these new data, it is intriguing how depletion of RNase MRP leads to quiescence. While it is possible that ribosomes fully assembled before RPP40 (and RNase MRP) depletion have long half-lives and are not turned over rapidly, the effects on proliferation from RPP40 down-regulation are rather immediate. The authors must describe these findings and provide a reasonable explanation, as this critical gap is related to the mainstay.

The relative levels of 28S and 18S rRNA (i.e., the 28S/18S ratios in Fig. S1K) indeed do not change upon depletion of RPP40. However, it is plausible that absolute cellular rRNA levels might be affected. While mature rRNA is stable in proliferating cells, it turns over with a half-life of 35-72 hours in non-proliferating animal cells (PMID 4500459; Abelson et al, Cell 1974, 161-165). To evaluate potential changes in total rRNA levels per cell, we quantified 28S rRNA levels in C40 cells over time using fluorescence *in situ* hybridization-flow cytometry (FISH-Flow; PMID 37481729). We observed a 20–30% reduction in 28S rRNA levels at 2 days and a 50–60% reduction at 7 days of dTAG treatment (Fig. R1; included as Fig. S1L-N).

Thus, although much of the cellular rRNA content remains intact at the onset of proliferative arrest between days 2 and 3 (Fig. 2A), rRNA levels are significantly reduced compared to vehicle-treated control cells (Fig. R1; Fig. S1L-N). We propose that this reduction contributes to the observed proliferative arrest. Additionally, independent of total rRNA levels, quiescent cells are known to sequester rRNA within functionally competent but inactive hibernating ribosomes (PMIDs 30476446, 22605777, 39379376, 32687489). It is conceivable that formation of such hibernating ribosomes plays a role in the proliferative arrest of C40 cells, irrespective of rRNA depletion. We have incorporated these results and comments in the revised manuscript (Fig. S1L-N; lines 147-150 and 398-407).

Figure R1 (new Fig. S1L-N). FISH-Flow analysis of 28S rRNA in dTAG-treated C40 cells. A-C Cells were treated with DMSO or dTAG for the indicated times, stained with DAPI and FISH probes targeting 28S rRNA, and analyzed by flow cytometry. Fluorescence intensities of stained or unstained (unst) cell populations are shown as histograms (A), contour plots (B), or as relative mean fluorescence intensities (MFIs) calculated by normalizing the geometric mean of 28S rRNA-Quasar670 intensity to DMSO-treated cells (C). Data in C are shown as mean \pm SD ($n = 4$). *, $p = 0.0225$; **, $p < 0.0005$ (one-way ANOVA followed by Dunnett's multiple comparisons test).

2. In many instances, it would be beneficial for the information provided in the rebuttal to be included in the revised manuscript (after addressing some additional perspectives listed below).

(a) It is evident that depletion of RPP21 and RPP40 leads to inactivation of RNase P in N21 and C40 cells, respectively. Moreover, there is a similar reduction in H1 RNA levels even though the protein subunits are affected rather differently. In the rebuttal and revised text, the authors mention that H1 RNA level would plummet if the holoenzyme were not fully assembled. This speculation is reasonable. However, the explanation provided for the lack of POP5 degradation seems inadequate. If contacts between POP5 and other RPPs in the finger module of RNase P account for protection of this protein but not RPP14, it is intriguing that the same argument does not hold in N21 cells. Overall, the interesting differences in the RPP levels in C40 versus N21 cells (Figure 1G versus Supplementary Figure 1I) would benefit from a stronger rationale.

Since POP5 is the only protein subunit of the palm module forming protein-protein interactions with the stable finger module, we speculate that these interactions confer the unique protection of POP5 but no other subunit of the palm module (including RPP14) from degradation. Identical interactions exist in C40 and N21 cells, so this speculation applies to both cell lines. However, we engineered C40 and N21 cells for inducible degradation of different subunits, RPP40 and RPP21, respectively. Whereas dTAG treatment of C40 cells induces degradation of the centrally positioned RPP40 (within RNase P and MRP complexes) along with many other protein subunits, dTAG treatment of N21 cells induces degradation of the peripherally located RPP21 (present only in RNase P) and no other protein subunits. Since RPP40 directly interacts with all subunits of the palm module as well as the RPP29 subunit of the wrist module (PMID 30454648), we envision RPP40 as an interaction hub whose depletion leads to disintegration of a large portion of the RNase P and MRP complexes with subsequent destabilization of several protein subunits. In contrast, RPP21 is exclusive to the wrist module and only interacts with two subunits, RPP29 and RPP38. Its occurrence at the periphery of RNase P and limited protein-protein contacts may explain why depletion of RPP21 preserves stability of all other protein subunits of RNase P. We clarified this speculation in the revised manuscript (lines 128-136).

(b) It is fine that the non-canonical pre-rRNA processing products accumulate when RNase MRP is depleted, but where do these products come from? Is it mis-cleavage by partially-assembled, weakly-active MRP? Dysregulation of specific endo/exonucleases? RNA self-cleavage?

The observed non-canonical pre-rRNAs processing products in RNase MRP-depleted cells most likely result from bypassing the ITS1 cleavage, which leads some precursor transcripts to be cleaved in ITS2 (a step that normally follows ITS1 cleavage), as proposed previously (PMIDs 23439679, 28115465, 22342225; Fig. 1I). We have included this comment in the legend to Fig. 1I in the revised manuscript. Because cleavages of the spacer sequences normally follow a strict 'timetable', presumably to keep the ribosome assembly line in check (PMID 22342225), it is plausible that these non-canonical pre-rRNAs, which are not processed in a timely fashion, are targeted for degradation.

(c) The slower moving bands in the H1 RNA northern blot are being compared to the scenario in the Akiyama et al. (2022) bioRxiv. However, this earlier paper only shows bands that migrate faster compared to the putative full-length H1 RNA species. Thus, Akiyama et al. (2022) bioRxiv does not seem an ideal reference. Perhaps, the slower RNA is a circularized version?

We agree that the faster-migrating bands detected by Akiyama et al. are distinct from the faint, slower migrating band seen above the full-length *RPPH1* RNA in our northern blot. We hypothesize that this slower band could represent a longer, primary transcript of *RPPH1* (most circular RNAs arise through splicing, which is unlikely in this case). We have included this comment in the legend to Fig. 1D.

(d) New data are provided with respect to tRNA transcription in the rebuttal (Fig R7) but not in the revised manuscript. The number of tRNA transcripts impacted in N21 cells, however, is significantly lower than in C40 cells. Is there any reason why this is the case? Regardless, please include this figure in the revised supplement and include a comment in the text.

We believe that the stronger perturbation of tRNA transcription in dTAG-treated C40 cells (in which both RNases P and MRP are depleted) compared to N21 cells (in which only RNase P is depleted) results from the potent regulatory effect of RNase MRP on cell metabolism, including transcription. We now show the analyses of tRNA transcription as Fig. S4L, M and we included the above comment in the figure legend.

(e) "Our results demonstrate a coherent transcriptional induction of genes associated with translation and tRNA processing in C40 but not N21 cells..." – Figure S4 shows there is still significant upregulation of translation and tRNA processing associated genes in N21 cells, at least at the 3-h time point, but not to the same extent as in C40 cells. At 24 h, these genes are significantly downregulated. Is there an explanation for reversal of this trend?

The expression profiles of genes associated with translation or tRNA processing in N21 cells at 3 h (Fig. S4J) or 24 h (Fig. S4K) does not substantially deviate from the expression profiles of all genes at the corresponding times in these cells (Fig. S4I). A slight deviation from the expected trend is the bias toward upregulation among the tRNA processing factors at 3 h (64 and 49 tRNA genes are significantly up- and downregulated, respectively, versus 5918 and 6907 up- and downregulated among all genes; Fig. S4I, J). We suspect that this may indicate a mild countering response to blocked tRNA biogenesis after depletion of RNase P in these cells, akin to the much stronger countering response in C40 cells to blocked rRNA biogenesis, as discussed in the manuscript.

3. To address a previous critique pertaining to the striking differences in accumulation of pre-tRNAs, the authors have repeated the northern blot for tRNA^{Arg} (new panel in Figure 1H, which is very different from the previous data). It is unclear if the loading control for this new blot is also being used as the reference for the older northern blots related to tRNA^{His} and tRNA^{Asn}. If so, the top two panels and their corresponding loading control (previous version of the manuscript) should be presented together, while the new tRNA^{Arg} blot should have its own loading control. Since the

loading control (SRNORD3A) data in the previous versus current submission are different, the revised Figure 1H appears to be a mélange. If so, please fix.

All northern blots shown in Fig. 1H in both the initial and revised versions of the manuscript used aliquots of the same total RNA samples. Therefore, we reasoned that presenting a single northern blot of *SNORD3A* as a loading control would be sufficient. We show the more recent *SNORD3A* northern blot of these RNA samples as it is of better quality than the initial one. We also analyzed biological replicates of this experiment, which yielded similar results.

4. In their rebuttal, the authors state “The faint band that appears at a higher-than-predicted molecular weight (MW) in the RPP20 blot in Fig. 2D is likely non-specific, while both bands in the RPP29 blots in Figs. 1G and 2F are likely specific, as their intensities change coherently.”. The latter comment does not seem to be accurate. In Figure 2F, it is obvious that the intensity of the two main RPP29 bands pre- and post-Dox induction are reversed. Perhaps, they constitute isoforms that arose from splicing or post-translational differences. Worth pointing out.

We agree and now highlight this possibility in the legend to Fig. 2F.

Minor points

1. Consider revising the title to “Cellular quiescence is induced upon rapid depletion of RNase MRP”.

Please see our introductory paragraph of this rebuttal. In response to the editorial request, we revised the title to “Reversible proliferative arrest induced by rapid depletion of RNase MRP”.

2. In response to the comment that the RPPs should be drawn to scale in Figure 1A, the authors stated that they followed the representation PMID 30454648. In Figure 5A of PMID 30454648, however, POP5 and RPP14 are similar in size. Therefore, for the sake of accuracy, the schematic in Figure 1A should be revised.

We revised the schematic of RNase P as shown in Figure R2 (new Fig. 1A).

Figure R2 (new Fig. 1A). Schematic of the human RNase P ribonucleoprotein, drawn based on the cryo-EM structure reported in Wu *et al.* (PMID 30454648). Protein subunits RPP25, RPP20, POP1, POP5, RPP30, RPP14, RPP40, RPP29, RPP21, RPP38 are colored according to their position in the finger, palm, or wrist modules. The catalytic RNA *RPPH1* is in gray. Asterisk indicates the RNase P-specific protein subunit, RPP21. Arrows indicate subunits that were targeted in this study.

3. Interesting that RNase MRP appears to localize to speckles in the N21 cells compared to the larger structures in C40 cells. Comment?

We found no significant difference in the size of *RMRP* FISH signals (i.e., nucleoli) between DMSO-treated N21 and C40 cells (Fig. R3A). We also realized that the larger FISH signals shown initially in Fig. 1F were not representative of their average size; we now use a more representative view in the revised Fig. 1F (Fig. R3B; see also Source Data file).

Figure R3. A No significant difference in the size of *RMRP* FISH signal area between N21 and C40 cells. Areas of at least 50 *RMRP* FISH signals were measured in fluorescent images of DMSO-treated C40 and N21 cells using ImageJ (version 1.54p). Data are shown as mean \pm SD. ns, not significant ($p = 0.682$; Student's *t* test). **B** Uncropped image of DMSO-treated C40 cells indicating the initial and current views (red dashed rectangles) shown in Fig. 1F. The uncropped image is shown in the Source Data file.

4. Could the authors speculate why S-phase arrest due to MRP depletion does not result in histone mRNA degradation as was observed when S-phase arrest is triggered by replication inhibition?

Degradation of replication-dependent histone mRNAs requires active translation (PMIDs 3028643, 16055702). While cells arrested in S-phase due to inhibition of DNA replication sustain high levels of translation and rapidly degrade their histone mRNAs, cells arrested due to RNase MRP depletion decrease their translation rate to less than 5% of the rate of control cells. It is thus likely that the dramatic reduction in active translation in S-phase cells lacking RNase MRP stabilizes histone mRNAs. We outline this possibility in lines 357-371 in the revised manuscript.

5. Pg 5, In 134: The data are not consistent with “a substantial accumulation”. Please revise text.
6. Pg 5, In 169: Revise to read “To validate the idea that.....”
7. Pg 5, In 172: Revised to read “...due to the slight...”
8. References: As indicated before, titles need to be uniformly formatted in sentence and not title case (including but not limited to references 2, 4, 18, 33, 58, 61).

Points 5–8: We addressed these points, as suggested.

Reviewer #2 (Remarks to the Author):

Reviewer #3 (Remarks to the Author):

I co-reviewed this manuscript with one of the reviewers who provided the listed reports. This is part of the Nature Communications initiative to facilitate training in peer review and to provide appropriate

recognition for Early Career Researchers who co-review manuscripts.

Reviewer #4 (Remarks to the Author):

In their revised version the authors have strengthened the manuscript by addressing the p53-dependency of the quiescence phenotype and by adding data using the RNA polymerase I inhibitor. Expanding the link between quiescence and rRNA biogenesis by depleting additional biogenesis factors would still have been important.

We agree that exploring additional rRNA biogenesis factors is an important direction for future research of quiescence, and we are actively pursuing this line of investigation.

To my opinion, the data on increased histone mRNA remain a weakness of manuscript. First, the data are somewhat disconnected from the rest of the manuscript. Second and more importantly, the functional connection of increased histone mRNA to quiescence as well as the mechanistic basis of the mRNA increase remain unclear.

We feel that these data are an important integral part of our study. Histone mRNAs represent the most significantly upregulated gene category during reversible proliferative arrest, directly linking them to the manuscript's broader theme. Their accumulation is particularly striking, as such transcripts are typically absent in arrested cells that are not synthesizing DNA.

To address the mechanistic basis of this increase, we demonstrate that histone mRNAs accumulate due to: 1) increased stability of normally processed transcripts, 2) a dramatic decline in translation, which prevents their degradation, and 3) sustained low level of transcription. While the precise function of the accumulated histone mRNAs remains unclear, we hypothesize that their storage may facilitate reentry of the arrested cells into the cell cycle. This provides a foundation for future studies on the reversibility of proliferative arrest, as discussed in the manuscript.

Reviewer #5 (Remarks to the Author):

Thank you for the response to my questions. Upon reading the reviews and the responses, I have a few additional questions:

1. Reviewer 1 asked whether there is a change in the levels of 18S and 28S rRNA in cells with RPP40 depletion. This seems like a critical question: is there less rRNA available for the cells, or just a change in rRNA biogenesis without an effect on rRNA levels?

We observe a consistent, time-dependent reduction in steady-state rRNA levels, indicating that RPP40 depletion leads to a net decrease in rRNA availability (Fig. S1L-N). Please see also our response to point 1 of Reviewer #1.

The authors responded that there is no change in relative abundance of 28S and 18S rRNA and state as data not shown that there was a modest reduction in total RNA from C40 cells on day 7.

Indeed, while the relative levels of 28S and 18S rRNA remain unchanged (Fig. S1K), we initially noticed that less total RNA could be extracted from C40 cells after 7 days of treatment with dTAG compared to DMSO-treated controls. Our FISH-Flow analysis now quantifies the per-cell levels of 28S rRNA, revealing a 20–30% reduction by day 2 and a 50–60% reduction by day 7 (Fig. S1L-N).

Do the cells have less rRNA? Could the effects reflect a scarcity of rRNAs? Or is there an impact of reducing rRNA biogenesis even if there is an abundant supply of rRNAs.

We speculate that both the progressive reduction in mature rRNA levels as well as defects in rRNA biogenesis contribute to the observed decline in critical cellular functions.

2. A related question is whether RPP40 depletion reduces translation rate. Even though the authors show this data, it is difficult to interpret because it is not clear whether reduced translation reflects reduced availability of rRNA or tRNA, or if it is a consequence of the induced cell cycle arrest.

Upon RPP40 depletion, the translation rate declines more rapidly than the availability of rRNA. For example, at 2 and 7 days of dTAG treatment, translation in C40 cells drops to below 50% and below 5% of the initial rate, respectively, outpacing the decline in 28S rRNA levels by approximately 2- and 10-fold (Fig. 3A, Supplementary Fig. 1N). This suggests that rRNA availability alone cannot fully explain the steep decline in translation. Although tRNA processing is also impaired, its contribution appears minimal, as depletion of RNase P alone does not significantly affect cell proliferation until day 6 of dTAG treatment (Fig. 1H, Supplementary Figs. 1O, 2A). We further consider it unlikely that the rapid translational decline results from the induced proliferative arrest. Instead, it is more plausible that the latter is a consequence of the reduced translation rate, as even a 50% reduction in translation (which we observe at day 2; Fig. 3A, B) is sufficient to induce cell cycle withdrawal (which occurs between days 2 and 3; Fig. 2A) (PMID 902318). Given that the rate of protein synthesis is limited by rRNA biogenesis (PMID 15688068), which is directly inhibited by depletion of RPP40, we propose that the primary driver of translational decline is disrupted rRNA biogenesis. We speculate that compromised rRNA biogenesis leads to translational decline through both reduced rRNA availability and signaling triggered by defects in rRNA processing. Further studies will be needed to dissect these mechanisms in greater detail. We have incorporated this comment in the revised manuscript (lines 396-411).